# Oxidation of F-actin controls the terminal steps of cytokinesis

Stéphane Frémont[1,2], Hussein Hammich[3], Jian Bai[1,2,4], Hugo Wioland[5], Kerstin Klinkert[1,2,4], Murielle Rocancourt[1,2], Carlos Kikuti[3], David Stroebel[6], Guillaume Romet-Lemonne[5], Olena Pylypenko[3], Anne Houdusse[3] & Arnaud Echard[1,2]

Cytokinetic abscission, the terminal step of cell division, crucially depends on the local constriction of ESCRT-III helices after cytoskeleton disassembly. While the microtubules of the intercellular bridge are cut by the ESCRT-associated enzyme Spastin, the mechanism that clears F-actin at the abscission site is unknown. Here we show that oxidation-mediated depolymerization of actin by the redox enzyme MICAL1 is key for ESCRT-III recruitment and successful abscission. MICAL1 is recruited to the abscission site by the Rab35 GTPase through a direct interaction with a flat three-helix domain found in MICAL1 C terminus. Mechanistically, in vitro assays on single actin filaments demonstrate that MICAL1 is activated by Rab35. Moreover, in our experimental conditions, MICAL1 does not act as a severing enzyme, as initially thought, but instead induces F-actin depolymerization from both ends. Our work reveals an unexpected role for oxidoreduction in triggering local actin depolymerization to control a fundamental step of cell division.

[1] Membrane Traffic and Cell Division Lab, Cell Biology and Infection Department Institut Pasteur, 25–28 rue du Dr Roux, 75724 Paris Cedex 15, France. [2] Centre National de la Recherche Scientifique UMR3691, 75015 Paris, France. [3] Structural Motility, Institut Curie, PSL Research University, CNRS, UMR 144, F-75005 Paris, France. [4] Sorbonne Universités, UPMC Univ Paris06, Sorbonne Universités, IFD, 4 Place Jussieu, 75252 Paris Cedex 15, France. [5] Institut Jacques Monod, CNRS, Université Paris Diderot, Université Sorbonne Paris Cité, 75013 Paris, France. [6] Ecole Normale Supérieure, PSL Research University, CNRS, INSERM, Institut de Biologie de l'École Normale Supérieure (IBENS), 75005 Paris, France. Correspondence and requests for materials should be addressed to A.E. (email: arnaud.echard@pasteur.fr).

Cytokinesis is the terminal step of cell division and leads to the physical separation of daughter cells. While cytokinesis is essential for cell proliferation, an important proportion of cancers likely result from a cytokinesis failure[1,2]. Cytokinesis starts in anaphase with large-scale deformation of the plasma membrane driven by a contractile ring made of actin and myosin II (ref. 3). This ring cannot lead to cell cleavage on its own, since for several hours the two daughter cells are connected by a microtubule-filled intercellular bridge, both in cultured cells and in vivo[4]. At the center of the bridge, the midbody or Flemming body serves as a platform for abscission. A major advance in the cell division field came from recognition that the Endosomal Sorting Complex Required for Transport (ESCRT), initially described for intraluminal vesicle formation in late endosomes and HIV budding, forms helices that locally pinch the plasma membrane and drive abscission[5–9]. Consistent with its key role in abscission, the ESCRT machinery is the target of the AuroraB-dependent NoCut checkpoint[9–12].

While F-actin and microtubules play pivotal roles in furrow ingression, these cytoskeleton elements must be cleared at the abscission site to allow constriction of the plasma membrane by the ESCRT machinery[3,4]. An important conceptual advance came upon discovery that the microtubule-depolymerizing enzyme Spastin is recruited by the ESCRT machinery in order to clear microtubules at the abscission site[13,14]. With respect to actin clearance, the small GTPases Rab35 and Rab11A function in parallel to prevent F-actin accumulation within the intercellular bridge[15,16]. Indeed, Rab35 recruits the Oculo-Cerebro-Renal syndrome of Lowe (OCRL) phosphatase to the intercellular bridge to locally hydrolyse PtdIns(4,5)P$_2$, a lipid that promotes actin polymerization in late cytokinetic bridges[17–19]. Similarly, Rab11A-endosomes transport p50RhoGAP that limits actin polymerization[20]. However, the mechanisms that actively depolymerize F-actin in the intercellular bridge, equivalent to Spastin for microtubules, remain to be discovered.

MICAL1, identified as a Molecule Interacting with CasL, belongs to the family of MICAL proteins, conserved from insects to humans[21]. MICALs are intracellular proteins that catalyse oxidation–reduction (redox) reactions through their flavoprotein monooxygenase (MO) domain and use F-actin as a substrate[22–25]. These enzymes bind flavin adenine dinucleotide (FAD)[26,27] and use the coenzyme nicotinamide adenine dinucleotide phosphate (NADPH) and O$_2$ in redox reactions, causing disassembly of F-actin likely by directly oxidizing specific actin methionines. Mechanistically, it was initially proposed that MICALs constitute a new family of actin severing enzymes[23] and a recent report indicates that MICAL-oxidized filaments are more efficiently severed by cofilin[28]. In Drosophila, MICAL-mediated actin remodelling regulates axon guidance and has essential roles in other actin-related processes such as myofilament organization, dendritic pruning and bristle development[22,29–31]. In Man, three MICAL genes (MICAL1, -2 and -3) have been identified and are required for fundamental biological processes, such as cell adhesion, cell migration, axon growth, angiogenesis, gene transcription and vesicle trafficking[25,32–36].

Despite the importance of MICALs in F-actin dynamics, nothing is known about their potential roles during cell division. Importantly, MICALs' enzymatic activity must be tightly regulated. For understanding how they locally remodel the actin cytoskeleton, a crucial question is to determine how these enzymes are activated and precisely targeted at specific cellular locations. Given that Rab35 controls actin dynamics in many cellular functions including cytokinesis[16] and that Rab35 interacts with MICAL1 by two-hybrid and co-immunoprecipitation (coIP)[36,37], it has been hypothesized that Rab35 together with MICAL1 might regulate cytokinesis[38].

Here we provide evidence for a conserved and fundamental role of MICAL1 in cytokinetic abscission, both in human and Drosophila cells. Active, GTP-bound Rab35 directly interacts with the tail of MICAL1 through a flat three-helix domain revealed by X-ray crystallography. This interaction is essential for localizing MICAL1 at the abscission site. Surprisingly, single filament assays using end or side-bound tethers demonstrate that the MICAL1-induced weakening of F-actin primarily induces rapid depolymerization from both ends rather than filament severing previously reported for both tethered[23] and unteathered[28] filaments. Importantly, we reveal that Rab35 activates the enzymatic activity of MICAL1 by displacing the inhibitory intramolecular interaction between the C-terminus of MICAL and its catalytic monooxygenase domain. Altogether, MICAL1 controls abscission by promoting F-actin depolymerization at the abscission site, which appears as a prerequisite for ESCRT recruitment at this location. Altogether, this work links oxidation with local cytoskeleton depolymerization, revealing an unexpected role for oxidoreduction in cell division.

## Results

**MICAL1 localizes at the abscission site.** During cell division, immunofluorescence on fixed samples revealed that cells expressing GFP-MICAL1 displayed cytoplasmic staining during prophase, metaphase and furrow ingression, but showed a distinct pattern during late cytokinetic stages (Fig. 1a; Supplementary Fig. 1a). Using β-tubulin and the ESCRT-III component CHMP4B as markers for the age of the intercellular bridges[7,8], MICAL1 was first found to accumulate on both sides of the midbody, where it colocalized with CHMP4B (Fig. 1a, red arrows point to midbodies). Of note, MICAL1 was not present at earlier stages, when CHMP4B was not yet present in the bridge (0 out of 57 bridges). Interestingly, while CHMP4B localization changes from the midbody to the adjacent abscission site, characterized by an interruption of the tubulin staining (secondary ingression site[7,8,20]), MICAL1 localization shifted from the midbody to a zone closely apposed to or at the abscission site (Fig. 1b, orange arrows point to abscission sites), where it partially colocalized with CHPM4B. Quantification revealed that 62% of the bridges ($n = 84$) displaying CHMP4B at the abscission site showed a detectable pool of MICAL1 apposed or at this location. The same results were obtained in CRISPR/Cas9-mediated genome-edited HeLa cell line that expressed MICAL1 tagged with GFP (GFP-MICAL1$^{endogenous}$) at the endogenous locus, ruling out overexpression artifacts (Fig. 1c; Supplementary Fig. 1b). Time-lapse fluorescent and phase-contrast microscopy further indicated that GFP-MICAL1$^{endogenous}$ is first recruited to the midbody, then accumulates transiently at the abscission site before abscission occurs (Fig. 1d). This suggests that MICAL1 might play a role during cytokinesis when ESCRT-III localization shifts from the midbody to the abscission site and before ESCRT-III-dependent helices constrict to completion.

We conclude that MICAL1 dynamically localizes to the intercellular bridge and overlaps with CHMP4B at the abscission site before cytokinetic abscission.

**MICAL1 is required for successful cytokinetic abscission.** To test whether MICAL1 has a role in cytokinetic abscission, HeLa cells were recorded for 48 h using phase contrast time-lapse microscopy to determine with a 10 min accuracy the timing of abscission in control versus MICAL1-depleted cells. Treatment with siRNA targeting MICAL1 led to at least a 90% reduction of endogenous MICAL1 protein levels (Fig. 2a). In control- and MICAL1-depleted cells, mitotic round up, furrow ingression and formation of the intercellular bridge occurred normally (Fig. 2b).

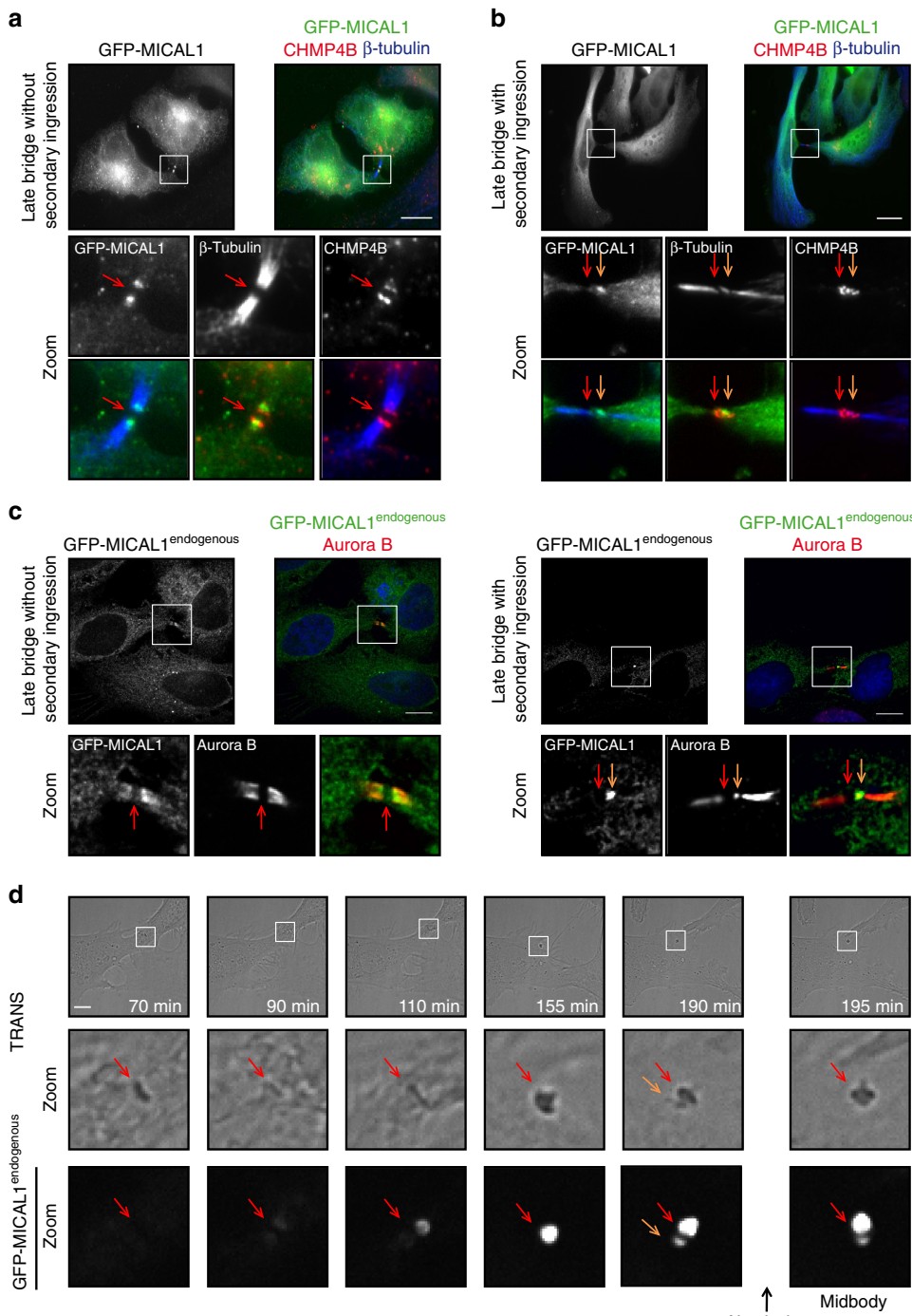

**Figure 1 | MICAL1 localizes in late intercellular bridges at the abscission site.** (**a,b**) HeLa transfected with GFP-MICAL1 (green) were stained with β–tubulin (blue) and CHMP4B (red). Scale bar, 10 μm. (**c**) Same as in **a** for a CRISPR/Cas9-edited HeLa cell line expressing GFP-MICAL1[endogenous]. Scale bar, 10 μm. (**d**) Snapshots of a time-lapse fluorescent and phase-contrast microscopy movie of the GFP-MICAL1[endogenous] genome-edited cell line. Scale bars, 10 μm. In all figures, red arrow: midbody and orange arrow: future abscission site.

In contrast, cytokinetic abscission was delayed after MICAL1 depletion (Fig. 2b,c, the two distributions are different with $P = 0.000$, Kolmogorov–Smirnov test). In particular, 20% of the MICAL1-depleted cells completed abscission >8.5 h after bridge formation, as compared to 3.5% in control cells. In addition, abscission was entirely blocked in 4.1% of divisions after MICAL1 depletion (more than threefold increase as compared to controls), and sister cells separate only mechanically by forces produced during rounding up in the following mitosis

(Supplementary Fig. 1c). Importantly, the abscission defects observed in MICAL1-depleted cells were completely rescued by the re-expression of a siRNA-resistant mRNA encoding GFP-tagged MICAL1, excluding off-target artifacts (Fig. 2d). Interestingly, expression of a redox-dead mutant of MICAL1 that is unable to oxidize F-actin (G90W-G92W-G95W, hereafter MICAL1[3G3W])[29] had a dominant negative effect and further delayed abscission (Fig. 2e), while it localized properly to the bridge (Supplementary Fig. 1d).

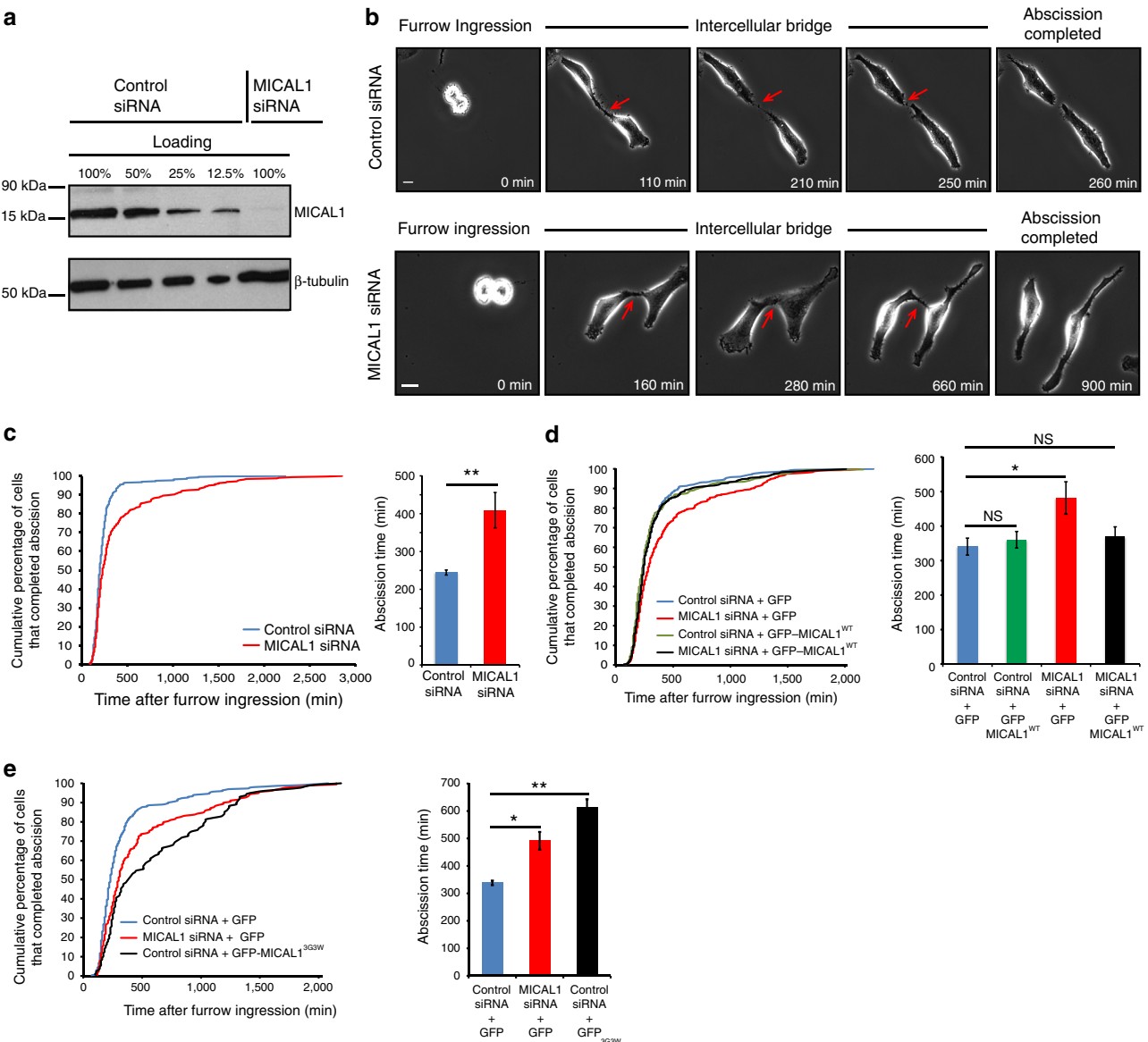

**Figure 2 | MICAL1 is required for successful cytokinetic abscission.** (**a**) Western blot showing MICAL1 depletion. Loading control: β-tubulin. (**b**) Snapshots of time-lapse phase-contrast microscopy movies of cells after control or MICAL1 depletion. Scale bars, 10 μm. (**c**) Distribution of the abscission times ($P = 0.000$, non-parametric and distribution-free Kolmogorov–Smirnov KS test) and mean abscission times (**$P = 0.006$, $t$-test) in control- and MICAL1-depleted cells ($N = 4$). $n = 316$-393 cells per condition. (**d**) Distribution of the abscission times and mean abscission times in control- and MICAL1-depleted cells transfected with the indicated plasmids ($N = 5$). No statistical differences between black, green and blue curves; $P = 0.000$ between red and other curves (KS test). NS, not significant; *$P < 0.05$; **$P < 0.01$ ($t$-test). $n = 320$–429 cells per condition. (**e**) Distribution of the abscission times and mean abscission times in control- and MICAL1-depleted cells transfected with the indicated plasmids ($N = 3$). $P < 0.008$ between blue and red or black curves (KS test). *$P < 0.05$; **$P < 0.01$ ($t$-test). $n = 128$–299 cells per condition. Error bars represent s.d.

MICAL proteins are conserved from Insects to Man and among the three human MICALs, MICAL1 has the most similar domain architecture to the unique *Drosophila* dMICAL protein[21]. To determine whether the function of MICAL in cytokinesis is evolutionarily conserved, we recorded *Drosophila* S2 cells upon dMICAL depletion using double-strand RNAs[39]. The timing of cytokinetic abscission was determined using Anillin-mCherry as a marker of the midbody, as previously described[40]. As in human cells, abscission was delayed in dMICAL-depleted cells indicating that MICAL is required for normal abscission in different species (Supplementary Fig. 1e).

These results reveal a conserved function of MICAL1 in cytokinesis and indicate that the redox enzymatic activity of MICAL1 is required for successful abscission.

**MICAL1 depletion leads to F-actin accumulation in bridges.** For fusion of the plasma membrane and final cut, all cytoskeletal elements, in particular F-actin, must be cleared at the abscission site[4]. Because MICALs have been shown to directly bind and disassemble actin filaments, we hypothesized that MICAL1 depletion might modify F-actin amounts at the intercellular bridge and thus impair cytokinetic abscission. Using fluorescent phalloidin as a marker for F-actin or a cell line expressing GFP-actin at endogenous levels, F-actin levels were found abnormally elevated in late intercellular bridges upon MICAL1 depletion (Fig. 3a; Supplementary Fig. 1f). Interestingly, this striking F-actin accumulation occurred at intercellular bridges rather than on cell bodies, suggesting a local action of MICAL1 during cytokinesis.

As the shape of the bridges looked irregular (Fig. 3a; Supplementary Fig. 1f), we investigated the effect of actin accumulation on 3D-bridge morphology using correlative light-scanning electron microscopy (SEM)[41,42]. In control cells expressing GFP-actin at endogenous levels, GFP-actin staining

was close to background levels by fluorescent microscopy and the corresponding membrane of the bridge appeared smooth by SEM (Fig. 3b, left). In contrast, GFP-actin levels were strongly and locally increased in the intercellular bridge in MICAL1-depleted cells, as expected (Fig. 3b, right). Strikingly, this was associated

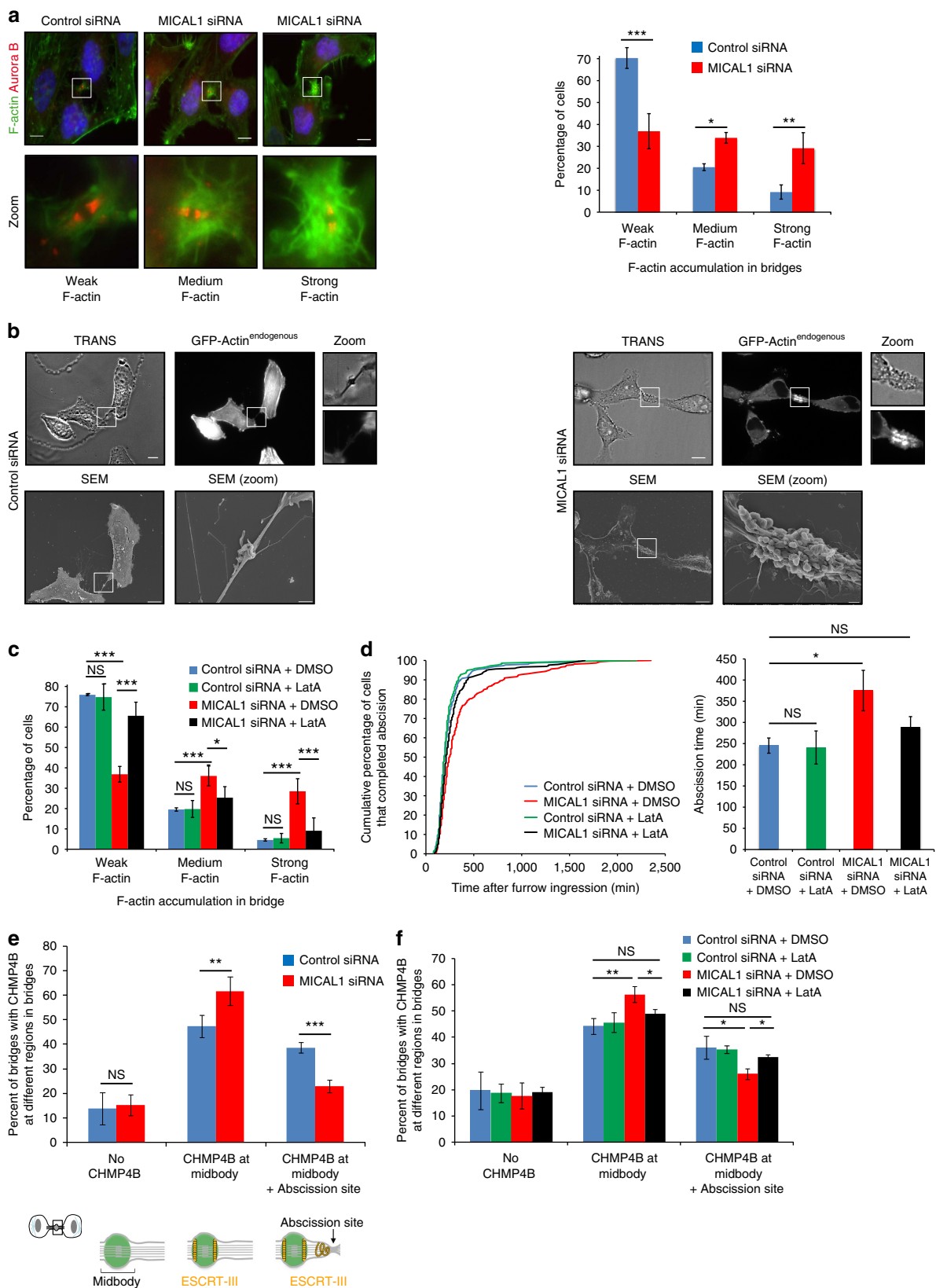

with a complete change in the bridge morphology, which displayed numerous extensions and blebs in the region around the midbody (16/19 bridges), a phenotype never observed in control cells (0/23 bridges). Membrane bulges and extensions were occasionally seen at the plasma membrane of the cell body, further indicating a locally restricted role of MICAL1 at the division site.

To directly investigate whether abnormal F-actin accumulation observed in MICAL1-depleted cells was the cause of the cytokinesis defects, cells were treated with very low, non-toxic amounts (20 nM) of the F-actin depolymerizing drug Latrunculin-A (LatA). We confirmed that the addition of 20 nM of LatA did not modify the levels of F-actin in control bridges (Fig. 3c, controls) and furthermore did not perturb cytokinetic abscission (Fig. 3d, controls). Interestingly, depolymerization induced by LatA treatment almost completely restored normal amounts of F-actin in intercellular bridges of MICAL1-depleted cells (Fig. 3c). This indicates that the extra amount of F-actin in these bridges is particularly sensitive to low doses of actin-depolymerizing drugs. Importantly, abscission defects associated with MICAL1 depletion were almost completely corrected by LatA treatment (Fig. 3d).

Altogether, these results demonstrate that F-actin accumulation in MICAL1-depleted cells is responsible for the observed defects in cytokinetic abscission.

**F-actin accumulation impairs CHMP4B recruitment.** Since F-actin accumulation inhibits cytokinetic abscission in MICAL1-depleted cells, we investigated whether ESCRT-III localization was defective. The ESCRT-III protein CHMP4B was present in ~85% of the cytokinetic bridges (either at the midbody only or at the midbody + the abscission site), both in control- and in MICAL1-depleted cells (Fig. 3e). However, CHMP4B was correctly recruited at the abscission site in only 23% of the bridges in MICAL1-depleted cells, in contrast to 39% in control cells (Fig. 3e). Concomitantly, an increase of bridges with CHMP4B localized only at the midbody was observed after MICAL1 depletion (Fig. 3e). Altogether, these results suggest that clearing F-actin from the bridge does not perturb initial ESCRT-III recruitment at the midbody but its subsequent recruitment at the abscission site.

To test whether F-actin accumulation underpinned defects in ESCRT-III localization, we treated control- and MICAL1-depleted cells with 20 nM of LatA. In line with the restoration of normal F-actin levels and normal abscission (Fig. 3c,d), the proportion of bridges with normal CHPM4B recruitment at the abscission site was partially, but significantly rescued by LatA treatment (Fig. 3f). These results mechanistically explain why normal abscission was observed after LatA treatment in MICAL1-depleted cells (Fig. 3d).

We thus conclude that the change in localization of ESCRT-III from the midbody to the abscission site critically depends on MICAL1 and requires a reduction of F-actin levels in the intercellular bridge.

**GTP-bound Rab35 directly interacts with MICAL1.** We next investigated by which mechanism MICAL1 is recruited to the abscission site during cytokinesis. Consistent with previous reports indicating potential direct interactions between MICALs and several Rab GTPases, including Rab35 (refs 36,37), we isolated clones encoding the C-terminal tail of MICAL1 (Fig. 4a) when we conducted a yeast 2-hybrid screen using the human GTP-locked mutant Rab35$^{Q67L}$ as bait and a human placenta complementary DNA (cDNA) library. Interestingly, Rab35 is an established regulator of actin remodelling during cytokinesis, both in *Drosophila* and human cells[16,17,43]. Yeast 2-hybrid experiments using truncated mutants of the C-terminal tail indicated that amino acids (aa) 879–1067 (MICAL1$^{879-1067}$) and aa 918–1067 (MICAL1$^{918-1067}$) interacted with GTP-bound Rab35$^{Q67L}$ and wild-type Rab35, but failed to interact with the GDP-bound Rab35$^{S22N}$ mutant (Fig. 4b; Supplementary Fig. 1g). Next, using recombinant proteins, the interaction between Rab35 and MICAL1$^{879-1067}$ was demonstrated to be direct and GTP-specific (Fig. 4c). This was confirmed by isothermal titration calorimetry (ITC), which determined that MICAL1$^{879-1067}$ and MICAL1$^{918-1067}$ fragments bound active, GppNHp-loaded form of Rab35 with comparable affinities ($K_d = 13$ and 6.4 μM, respectively), whereas no measurable binding was found with GDP-Rab35 (Supplementary Table 1 and Supplementary Fig. 2). We also determined that the last 26 aa of Rab35 were not necessary for the interaction (Supplementary Fig. 2). Finally, endogenous MICAL1 was preferentially immunoprecipitated by Flag-tagged GTP-bound mutant Rab35$^{Q67L}$ (Fig. 4d), suggesting that MICAL1 is a bona fide effector of Rab35.

As expected if Rab35 and MICAL1 function together, mCherry-MICAL1 colocalized in the intercellular bridge with GFP-Rab35$^{endogenous}$ (Rab35 tagged at the endogenous locus in TALEN-edited cells[19]), both on the lateral parts of the midbody and later at the abscission site (Fig. 4e). When cells expressed the dominant negative Rab35$^{S22N}$ mutant to inhibit Rab35 activation[43,44], a decrease in the number of bridges displaying detectable levels of MICAL1 was observed (Fig. 4f). This is consistent with MICAL1 being an effector of Rab35 during cytokinesis, and indicates that active, GTP-bound Rab35 contributes to MICAL1 recruitment to the abscission site.

In order to find a mutant of MICAL1 that has lost its ability to interact with Rab35, we screened deletion mutants in the MICAL1 C-terminal tail by two-hybrid assays. Whereas the C-terminal tail of MICAL1 (MICAL1$^{879-1067}$) interacted with

**Figure 3 | MICAL1 depletion leads to F-actin accumulation in cytokinetic bridges associated with abnormal CHMP4B recruitment and abscission defects.** (**a**) Left: Staining of F-actin by phalloidin (green), AuroraB (red) and DAPI (blue) in control- or MICAL1-depleted cells. Right: quantification of the intensity of the phalloidin staining in bridges after control and MICAL1 depletion ($N = 4$). *$P < 0.05$; **$P < 0.01$; ***$P < 0.001$ (two-way ANOVA). $n = 321$–365 cells per condition. Scale bars, 10 μm. (**b**) Correlative light-scanning electron microscopy (SEM) of dividing cells expressing GFP-actin$^{endogenous}$ after control (left) or MICAL1 (right) depletion. Phase contrast (TRANS), fluorescent and SEM pictures with corresponding zooms of cytokinetic bridges are presented. Scale bars, 10 μm (except for SEM zooms: 1 μm). (**c**) Quantification of the phalloidin staining in bridges in control- or MICAL1-depleted cells treated with DMSO or 20 nM Latrunculin-A (LatA) ($N = 3$). NS, not significant; *$P < 0.05$; ***$P < 0.001$ (two-way ANOVA). $n = 152$–194 cells per condition. (**d**) Distribution of the abscission times and mean abscission times in control- and MICAL1-depleted cells treated with either DMSO or Latrunculin A, as indicated ($N = 3$). No statistical differences between black, green and blue curves; $P < 0.002$ between red and other curves (KS test). NS, not significant; *$P < 0.05$; ***$P < 0.001$ (two-way ANOVA). $n = 218$–321 cells per condition. (**e**) Percentage of bridges with no CHMP4B at all, with CHMP4B only at the midbody and with CHMP4B at the midbody + at the abscission site in control− or MICAL1-depleted cells ($N = 4$). **$P < 0.01$; ***$P < 0.001$ (two-way ANOVA). $n = 288$–320 cells per condition. The abscission site is defined as an interrupted tubulin staining with a spot of CHMP4B, on one side of the midbody. (**f**) Same as in **e** after treatment with either DMSO or Latrunculin-A, as indicated ($N = 3$). NS, not significant; *$P < 0.05$; **$P < 0.01$ (two-way ANOVA). $n = 152$–212 cells per condition. Error bars represent s.d.

Rab35$^{Q67L}$, removing the last 41 aa (MICAL1$^{879-1026}$) completely ablated interaction with the active form of Rab35 (Fig. 4g). Direct binding and nucleotide specificity were confirmed using *in vitro* binding assays (Fig. 4c), as well as ITC experiments (Supple-mentary Table 1; Supplementary Fig. 2). Consistently, ablating interaction with Rab35 by deleting the last 41 aa of full-length MICAL1 (MICAL1$^{1-1026}$) totally abolished MICAL1 locali-zation at the bridge (0 out of 116 bridges; Fig. 4h). In parallel,

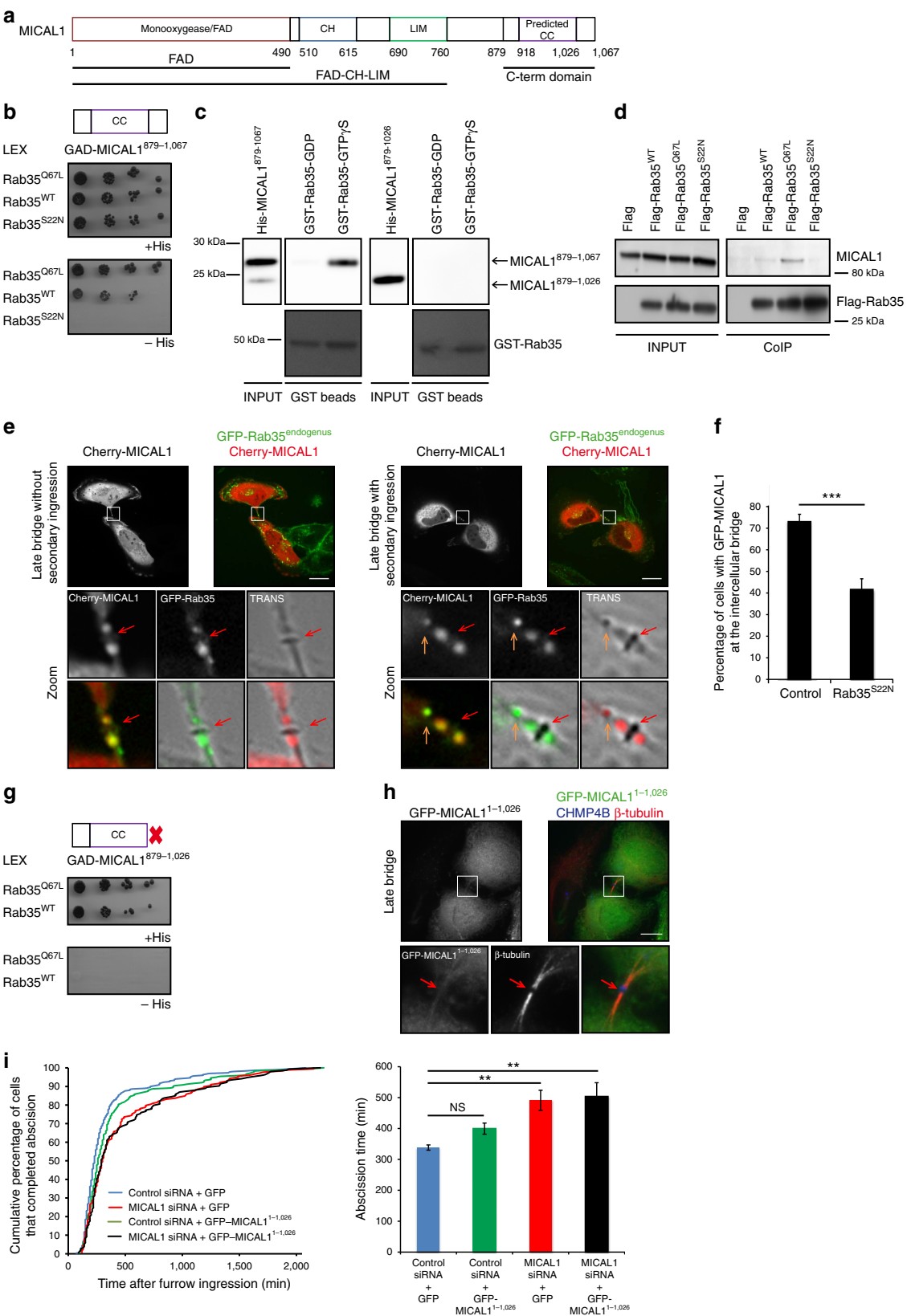

MICAL1[1–1026] lacking the last 41 aa failed to rescue abscission defects upon MICAL1 depletion (Fig. 4i), in contrast to full-length MICAL1 (Fig. 2d).

We conclude that the Rab-binding domain is crucial for MICAL1 recruitment to the abscission site and for MICAL1 function during cytokinesis.

**The tail of MICAL1 adopts a flat three-helix domain**. The interaction with Rab35 required the last 150 residues of MICAL1 (aa 918–1067), in which the last 41 residues are essential (Fig. 4c,g). The C-terminal region of MICAL1 has been described as comprising segments of coiled-coils[21,37], potentially implying oligomerization of this enzyme[21,37]. We determined a structure of this domain by X-ray crystallography at 3.3 Å resolution (Fig. 5a; Supplementary Fig. 3a; Supplementary Table 2). Surprisingly, the structure consists of a curved sheet of three helices[45], exposing two opposite flat surfaces (Fig. 5a), which differs from most three helices folds that usually form compact bundles (Supplementary Fig. 3b). This structure is formed by anti-parallel intramolecular coiled-coils interactions: while the first part of helix H2 makes coiled-coil interactions with helix H1, the second part of helix H2 interacts with helix H3 also with coiled-coil interactions, and no interactions are formed between the H1 and H3 helices (Fig. 5a, Supplementary Fig. 3b). Biophysical measurements by MALS and SAXS indicated that this domain is a monomer (Supplementary Figs 4 and 5), and thus does not form an elongated dimeric coiled-coil, as previously found for other Rab-effectors[46,47]. The residues stabilizing inter-helical interactions are conserved for MICAL family members (Supplementary Fig. 4), except MICAL2 that does not possess this C-terminal domain. Actually, the C-terminal domain fold is conserved for other members of the MICAL family, as it has been recently demonstrated for MICAL-cL, MICAL1 and MICAL3 (ref. 48). In fact, our structure superimpose with each of these structures with root-mean-square deviation (r.m.s.d.) of 1.1 Å for 101 residues. Overall, the domain of ~80 × 30 Å in dimension exposes two large curved surfaces on either side of the helices (Fig. 5a).

To define how Rab35 could bind to MICAL1, we generated two different series of mutations (S1 and S2) by selecting exposed and conserved surface residues on opposite surfaces of the three-helix sheet (Fig. 5a; Supplementary Fig. 4: yellow residues for mutant S1 (E946K, V950D, E953K, V971E, L975R and V978E) and orange residues for mutant S2 (R1012E, M1015R, L1034K and V1038E)). We characterized their ability to bind to Rab35 by yeast 2-hybrid assays (Supplementary Fig. 6a–c) and ITC (Supplementary Table 1; Supplementary Fig. 2), and we demonstrated by SAXS their ability to conserve the native fold despite the mutations (Supplementary Fig. 5). These mutants allowed us to delineate the surface responsible for Rab35 binding since both WT and the mutant S1 were able to bind Rab35 with

similar affinity and a stoichiometry of 1:1, while the mutant S2 had lost all ability to bind Rab35 (Supplementary Table 1). The mutant S2 (R1012E, M1015R, L1034K and V1038E) and the single mutant M1015R had very low binding affinity to Rab35 indicating that both H2 and H3 helices participate in Rab35 binding (Supplementary Table 1; Supplementary Fig. 6a–c), while the opposite face of the three helical domain defined by the S1 mutations do not participate in Rab35 binding. To further delineate this Rab-binding interface, we probed the region found at the opposite end of the H3 helix with two single mutations I1048R and R1055E (Fig. 5a; Supplementary Fig. 4, red residues). While Rab35 binding was abolished for R1055E and significantly reduced for I1048R (Supplementary Table 1; Supplementary Fig. 6a–c), the E1001R mutation on the other side of this surface (Fig. 5b,c) showed that this conserved helix H2 residue is not essential for Rab35 binding by 2-hybrid assays (Supplementary Fig. 6c). The surface surrounding the critical Rab35 binding residues (Fig. 5b) is in large part composed of exposed hydrophobic residues surrounded by charged residues (Fig. 5c). Most of these residues are conserved among MICAL proteins (Supplementary Fig. 4), consistent with Rab-binding properties of several MICAL family members[37,49] and with the crystal structure of Rab1 bound to MICAL-cL[48]. This Rab35-binding surface (oval in Fig. 5c) explains why the Δ41 fragment (aa 918–1026), which lacks the whole H3 helix, is unable to bind Rab35 (see above). Consistent with helices H2 and H3 but not H1 being crucial for Rab35 binding, introduction of the single mutations M1015R, I1048R or R1055E in full-length MICAL1 prevented MICAL1 localization to cytokinetic bridges (Fig. 5d), while mutant S1 still localized properly (Supplementary Fig. 6d). In conclusion, we identified key residues involved in Rab35 binding and thus required for MICAL1 localization during cytokinesis.

**Rab35 activates the enzymatic activity of MICAL1**. Several experiments demonstrated that the catalytic monooxygenase/FAD domain of MICAL1 (MICAL1[1–499] or 'FAD' hereafter, Fig. 4a) disassembles actin filament *in vitro* using bulk assays, but how MICAL induces oxidation-mediated actin disassembly remains poorly understood[23,24]. Since the purified catalytic domain of *Drosophila* MICAL was reported to fragment individual actin filaments attached to a coverslip, MICAL1 has been proposed as a novel F-actin severing enzyme[23]. We reinvestigated in more detail the mechanism of actin disassembly by human MICAL1 using microfluidics, where the filaments are anchored by their stabilized pointed end only, while the dynamics of their free sides and free barbed end can be monitored accurately[50] (Fig. 6a). The introduction of MICAL1 FAD domain dramatically increased the depolymerization rate of actin filaments in a NADPH-dependent manner (Fig. 6b). Quantifications revealed that the barbed end depolymerized at a rate of up to 50 actin

**Figure 4 | GTP-bound Rab35 directly interacts with MICAL1 and contributes to its localization at the intercellular bridge.** (**a**) Domains of MICAL1. CH, calponin homology; LIM, Lin1, Isl-1 and Mec3; CC, predicted coiled-coil. (**b**) *S. cerevisiae* reporter strain expressing indicated GAD- and LEX fusion proteins, and grown on selective medium with or without Histidine. (**c**) Recombinant GST-tagged wild-type Rab35 proteins loaded with either GDP or GTPγS were incubated with recombinant His-tagged MICAL1[879–1067] or MICAL1[879–1026]. Western blot anti-6xHis and Ponceau S red staining are presented. Input: 1%. (**d**) Flag-tagged proteins from HEK 293 T cells transfected with the indicated plasmids were immunoprecipitated and revealed with anti-Flag antibodies (Input: 4%). Endogenous co-immunoprecipitated MICAL1 is revealed with anti-MICAL1 antibodies. (**e**) A TALEN-edited HeLa cell line expressing GFP-Rab35[endogenous] was transfected with plasmids encoding mCherry-MICAL1. Fluorescent and phase contrast pictures are displayed. Scale bars, 10 μm. (**f**) Percentage of cells with GFP-MICAL1 at the intercellular bridge in control- or Rab35[S22N]-expressing cells (N = 3). ***P < 0.001 ($\chi^2$-tests). n = 256–170 cells per condition. (**g**) Same as in **b** for the indicated GAD- and LEX fusion proteins. (**h**) Cells transfected with GFP-MICAL1[1–1026] truncated mutant (green) were stained with β − tubulin (red) and CHMP4B (blue). Scale bar, 10 μm. (**i**) Distribution of the abscission times and mean abscission times in control- and MICAL1-depleted cells transfected with the indicated plasmids (N = 3). No statistical differences between red and black curves; P < 0.032 between blue and green and P = 0.000 between red or black and other curves (KS test). NS, not significant; **P < 0.01 (t-test). n = 154–299 cells per condition. Error bars represent s.d. Red arrow: midbody and orange arrow: future abscission site.

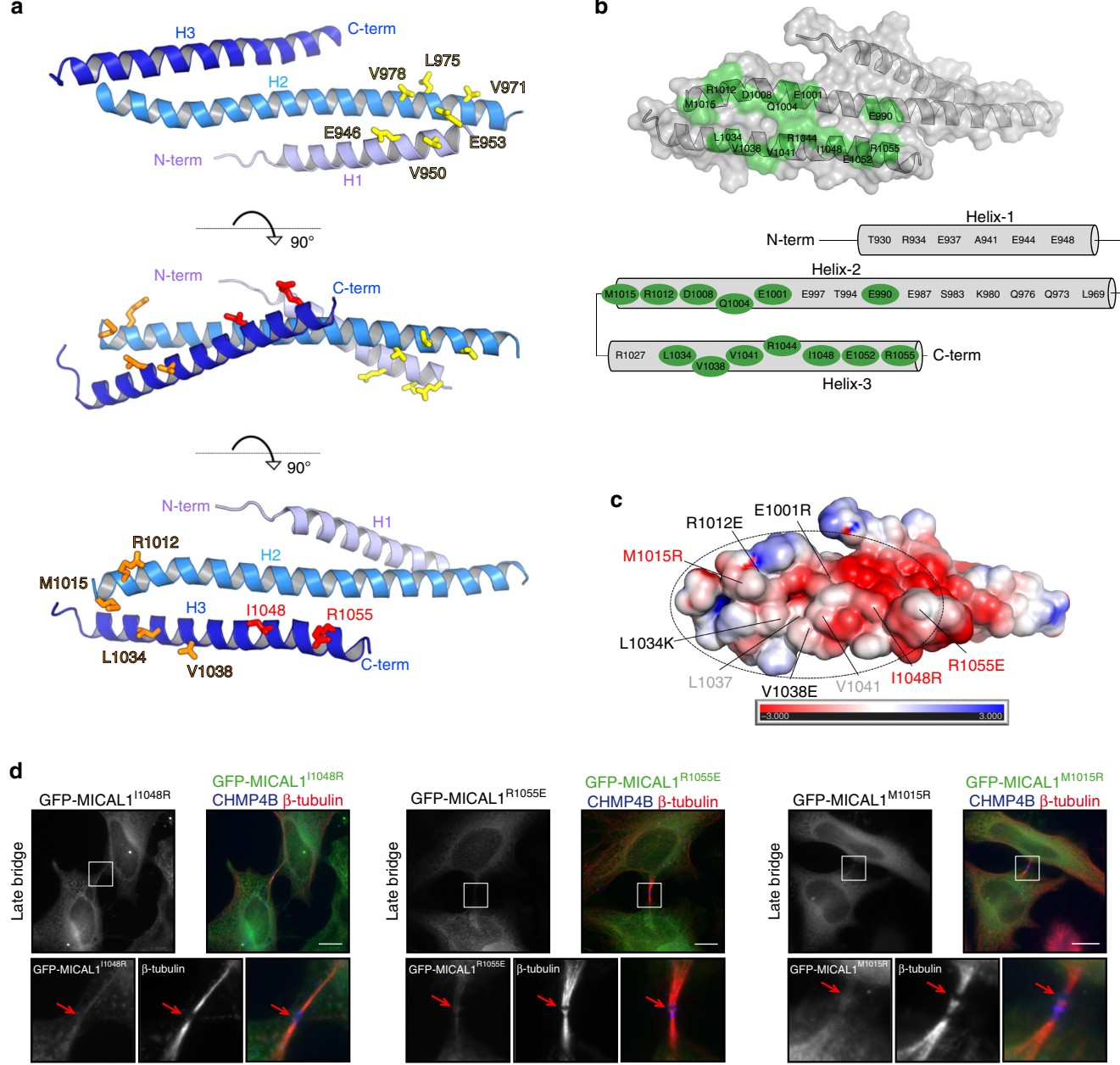

**Figure 5 | MICAL1 interacts with Rab35 through a flat three-helix domain. (a)** Structure of human MICAL1[918-1067] C-terminal domain involved in Rab35 binding. It consists of three amphipathic α-helices (H1, H2 and H3) connected by disordered mobile loops not visible in the structure. Two different sets of conserved and exposed residues were identified on either side of the flat α-helical structure as candidates for making direct contacts with active Rab35: Surface 1 in yellow (S1: E946, V950, E953, V971, L975 and V978) and Surface 2 in orange (S2: R1012, M1015, L1034 and V1038). The side chains of aa I1048 and R1055 essential for Rab35 interaction are displayed in red. **(b)** Conserved residues of MICAL and MICAL-like C-terminal domains mapped on the surface of MICAL1 H2 and H3 (see also Supplementary Fig. 4). A schematic model is also shown. **(c)** Electrostatic potential surface representation (contoured at ± 3 kT/eV; blue/red) of the C-terminal domain of MICAL1, as calculated with APBS[69,70] and visualized with Pymol. Single mutations that abolish Rab35 binding (red labels), set of mutants involved in Rab35 binding (black labels) and E1001R mutant that is not involved in Rab35 binding (white label) are indicated. Hydrophobic residues in the central part of the potential Rab35 interaction site are labelled in grey. **(d)** HeLa cells transfected with GFP-MICAL1 with indicated point mutations (green) were stained with β-tubulin (red) and CHMP4B (blue). Scale bars, 10 μm. Red arrow: midbody.

subunits per s in the presence of FAD and NADPH, more than eightfold faster than control filaments, which depolymerized at a rate of $6.0 \pm 0.7$ subunits per s (Fig. 6b, $N = 50$ filaments). The rate of F-actin depolymerization increased over time and reached a plateau ∼150 s after exposure to FAD (Fig. 6b). Interestingly, rapid depolymerization continued after removal of FAD from the microfluidic chamber (Fig. 6c, left), showing that oxidized actin filaments depolymerize faster regardless of the presence of

MICAL1 in solution. To further consolidate this observation, actin filaments were exposed to FAD in the presence of NADPH, the enzyme was removed from the microfluidic chamber and re-polymerization of the same filaments with fresh, non-oxidized actin monomers was achieved. After removal of actin monomers, filaments slowly depolymerized until the barbed end reached the oxidized lattice, where it started to quickly depolymerize even though FAD was absent from the solution (Fig. 6c, right). We

conclude that actin subunits oxidized by MICAL1 depart more rapidly from the barbed end.

Unexpectedly, we did not detect any increase in fragmentation events when exposing filaments to FAD in our microfluidics experiments. This result contrasts with a previous report, where

filaments were anchored to the coverslip by inactivated myosins and observed with Total Internal Reflection Fluorescence (TIRF) microscopy[23]. We thus repeated our experiments using these conditions, and consistently observed an acceleration of depolymerization from filament ends but no severing of filaments by

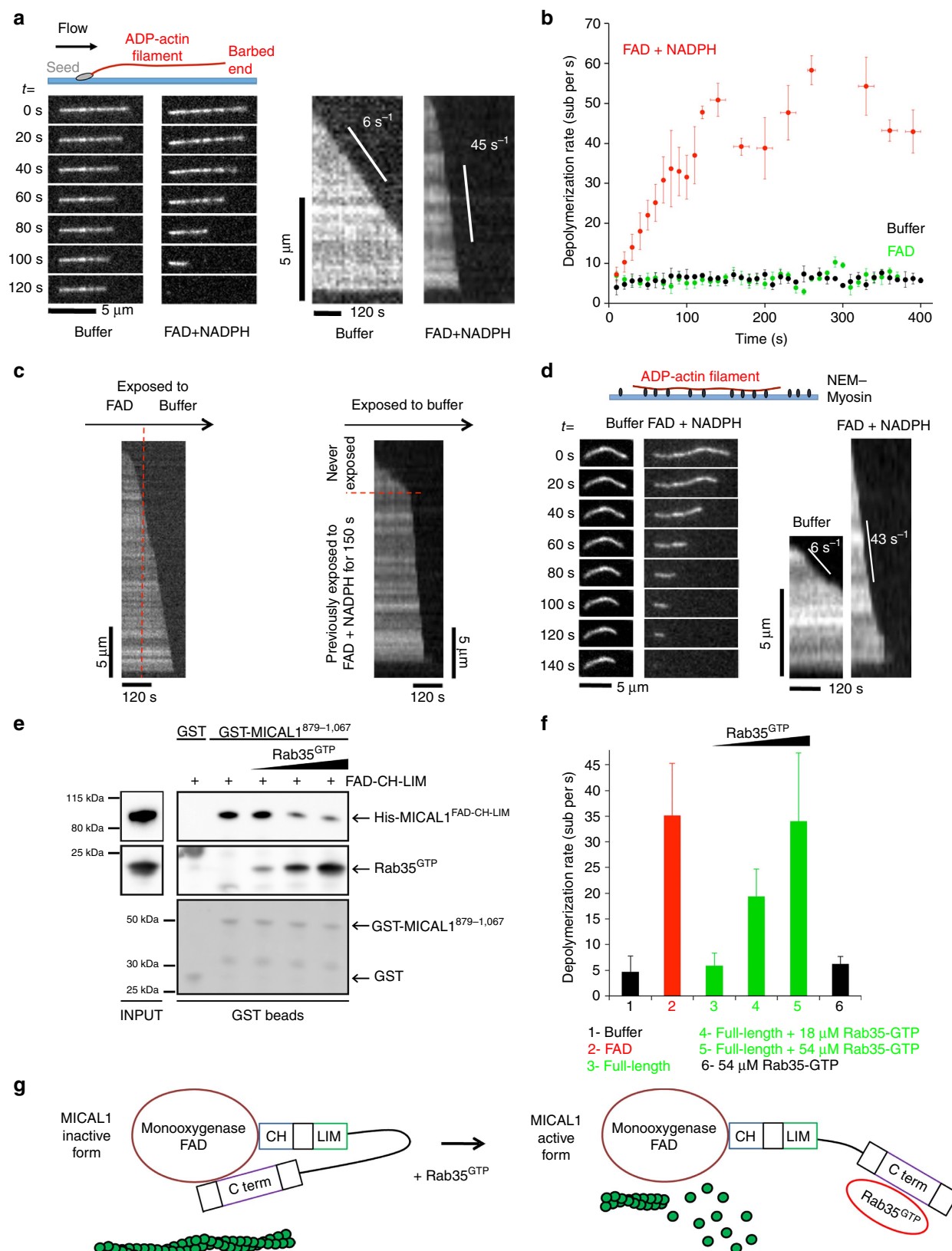

FAD (Fig. 6d). We conclude that MICAL1 destabilizes the whole filament lattice and accelerates its depolymerization from the barbed ends, instead of severing actin filaments. Notably, we also observed that FAD enhanced pointed end depolymerization, in both setups. We quantified the pointed end dynamics of filaments anchored to the surface by their barbed ends in our microfluidics set-up, and found that FAD increased pointed end depolymerization more than eightfold as well, from $0.17 \pm 0.07$ subunits per s in controls to $1.44 \pm 0.34$ subunits per s (s.d., $N = 25$ filaments). Altogether, we conclude that MICAL1-induced oxidation weakens subunit interactions within the actin filament, which enhances depolymerization from both ends.

One key question regarding this new family of actin-depolymerizing enzymes is how they are activated at the right place and time. Several studies demonstrated that the full-length protein is catalytically inactive and that the C-terminal extremity somehow inhibits enzyme activity[33,51]. Indeed, overexpression of MICAL1 does not disassemble the cellular actin cytoskeleton, unless the C-terminal extremity is truncated or mutated. We first determined using recombinant proteins that MICAL1[1–843] (hereafter 'FAD-CH-LIM', see Fig. 4a) directly interacted with the last C-terminal third of the protein MICAL[879–1067] (Fig. 6e), consistent with an intramolecular folded conformation for MICAL1. Demonstrating that the folding is inhibitory, incubation with increasing amounts of MICAL[879–1067] progressively inhibited the depolymerizing activity of FAD-CH-LIM (Supplementary Fig. 7a,b). Remarkably, the addition of Rab35 loaded with GTP displaced the interaction between the C-terminal domain and FAD-CH-LIM, suggesting that binding of Rab35 regulates enzyme activity (Fig. 6e). Confirming this hypothesis the depolymerizing activity of full-length MICAL1 was greatly enhanced in the presence of active Rab35, reaching the same depolymerization rates as when filaments were exposed to the non-inhibited FAD domain (Fig. 6f,g).

These results demonstrate that MICAL1-dependent actin oxidation induces depolymerization of F-actin filaments from both ends. In addition, Rab35 binding to MICAL1 fully releases the inhibitory interaction between the enzymatic and the C-terminal domains. This provides an original mechanism of activation of MICAL proteins by Rab GTPases.

## Discussion

Successful abscission requires that all cytoskeletal elements are removed from the abscission site to allow ESCRT-III helices to pinch the plasma membrane between the two daughter cells. While microtubules are depolymerized by the ESCRT-associated enzyme Spastin, the equivalent machinery that depolymerizes F-actin remains elusive. Here we identified MICAL1 as a critical enzyme localized in late cytokinetic bridges that promotes F-actin clearance at the abscission site (Fig. 7).

In the absence of MICAL1, F-actin accumulates in intracellular bridges, ESCRT-III components do not localize to the abscission site properly and cytokinetic abscission is delayed. The delay is even increased when a catalytically dead mutant MICAL1[3G3W] is expressed, highlighting the critical role of MICAL1 in abscission by controlling actin depolymerization through oxidoreduction. Importantly, we found that MICAL1 plays an evolutionarily conserved role in abscission from *Drosophila* to human cells. Intriguingly, the MICAL family members MICAL-L1 (MICAL-like 1) and MICAL3 also play a role in cell division[52,53]. However, this must be by a different mechanism, since MICAL3 functions in cytokinesis independently from its redox domain and MICAL-L1 lacks this enzymatic domain. Actually both MICAL-L1 and MICAL3 function in cell division by acting as scaffold proteins interacting with Rab11- and Rab8A-positive vesicles. Thus, MICAL-L1 and MICAL3 (through membrane trafficking)[49,52–54] as well as MICAL1 (through F-actin depolymerization, this study), play critical yet distinct roles in cytokinesis.

Our results reveal a new important mechanism that controls the timing of abscission, since they demonstrate that actin depolymerization is a prerequisite for proper ESCRT-III localization at the abscission site, but not earlier at the midbody/Flemming body (Fig. 7). Interestingly, Jasplakinolide treatment reduced ESCRT-III recruitment to the abscission site (Supplementary Fig. 8a), suggesting that chemical stabilization of actin also perturbs the recruitment of the abscission machinery. One possibility is that local actin depolymerization at, or close to the future abscission site regulates membrane tension within the intercellular bridge, which has been proposed to drive the translocation of ESCRT-III components from the Flemming body to the abscission site[55]. Alternatively, F-actin might represent a physical barrier that does not permit the recruitment of ESCRT-III at the abscission site[4]. Interestingly, the presence of the Arp2/3 subunit p34[Arc] in MICAL1-depleted bridges (Supplementary Fig. 8b) suggests that branched actin networks actually accumulate in these bridges. Of note, the fact that ∼50% of the cells undergo abscission with normal timing after MICAL1 depletion suggest that additional mechanisms exist in order to clear actin from cytokinetic bridges. For instance, Rab35 together with the PtdIns(4,5)P$_2$ phosphatase OCRL and Rab11 together with p50RhoGAP both limit actin polymerization in cytokinetic bridges[17,20], and could act in a redundant manner with MICAL1. Altogether, Rab35 controls F-actin levels during abscission by both limiting actin polymerization through OCRL and actively promoting its depolymerization through MICAL1.

How MICAL1 localization is determined in cells was poorly understood. Here we identified a tight functional link between MICAL1 and the Rab35 GTPase during cytokinesis.

**Figure 6 | MICAL1 is activated upon Rab35 binding and markedly accelerates actin filament depolymerization from both ends though oxidation. (a)** In a microfluidics set-up, ADP-actin filaments grown from surface-anchored seeds align with the flow. The fluorescent images show the typical barbed end depolymerization of individual filaments exposed to buffer alone (left) or to a solution of 600 nM FAD + 120 μM NADPH (right). The kymographs correspond to the same two filaments. **(b)** Depolymerization velocity measured over time for ADP-actin filaments exposed either to buffer alone (black), to 600 nM FAD (green) or to 600 nM FAD + 120 μM NADPH (red). Each set of data corresponds to a different population of 50 filaments observed in the microfluidics set-up. Data points are averages over the different filaments, and error bars are s.d.'s. **(c)** Kymographs of filaments depolymerizing in the microfluidics set-up, which is used to rapidly change the flowing solution to which the filaments are exposed. Left: an ADP-actin filament is exposed to 600 nM FAD + 120 μM NADPH for 85 s, followed by buffer alone. Right: An ADP-actin filament exposed to 600 nM FAD + 120 μM NADPH for 150 s was regrown from fresh (unoxidized) actin and depolymerized in buffer alone. **(d)** ADP-actin filaments are anchored to a surface coated with inactive myosins and are exposed to the same depolymerizing solutions as in **a**. **(e)** Recombinant GST-tagged MICAL1[879–1067] or GST alone were incubated with recombinant His-tagged MICAL1[FAD-CH-LIM] and increasing amounts of active Rab35-GppNHp. Western blot anti-His$_6$, anti-Rab35 and Ponceau S red staining are presented. Input: 1%. **(f)** Depolymerization rates measured on surface-anchored ADP-actin filaments with different solutions (6–31 filaments per condition). All solutions contain 120 μM NADPH. Bars show mean ± s.d. Concentrations of FAD and MICAL1 full-length were 1.2 μM. **(g)** Model for activation of the redox enzyme MICAL1 by Rab35-GTP (red). Actin is in green.

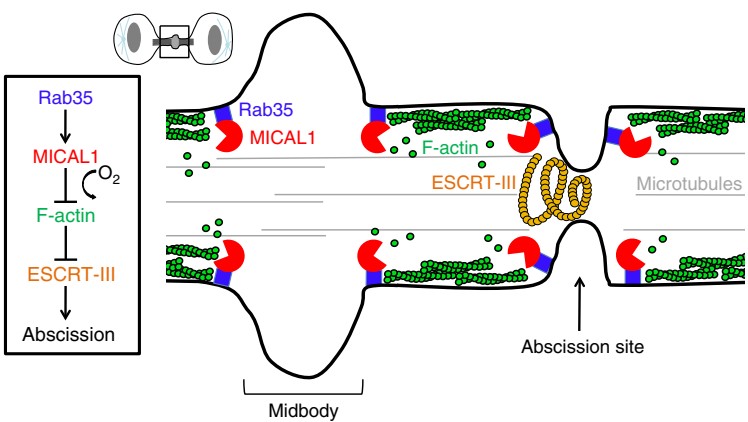

**Figure 7 | Model for F-actin clearance by oxidation during cytokinetic abscission.** Rab35 (blue) recruits and activates the redox enzyme MICAL1 (red), which directly depolymerizes F-actin (green) through oxidation in the intercellular bridge. This step is essential for normal ESCRT-III (orange) recruitment at the ingression site and for successful abscission.

Mechanistically, we demonstrate that GTP-bound Rab35 directly interacts with the C-terminal domain of MICAL1 and that Rab35 contributes to MICAL1 recruitment to the abscission site. Members of the MICAL and MICAL-like families have been reported to interact via a predicted coiled-coil domain with various Rab GTPases[21,37,49,54], reminiscent of other Rab/coiled-coil domain interactions with parallel dimeric coiled-coil proteins[46,47]. However, the structure of the Rab binding domain (RBD) of MICAL1 that we determined and several structures recently determined for Rab/RBD complexes[48] show that the nature of the interactions between MICALs and Rab proteins are quite different, since the Rab-interacting domain of MICAL1 folds into a structural domain composed of a sheet of three anti-parallel helices[45]. Based on single-mutant analysis, we predict that the H2 and H3 helices are critical for Rab35 binding. Two potential Rab binding sites of MICAL proteins have recently been identified with the structure of the MICAL1 RBD bound to two Rab10 molecules (5LPN)[48]. In light of this study, our mutational analysis indicate that Rab35 binds to only one of the two Rab binding sites described for MICAL1 RBD, as does Rab1 (ref. 48). Consistently, only the Rab binding site involving the H2 and H3 helices plays a role for MICAL1 recruitment to the bridge.

An unexpected result is the finding that, in our experiments under different conditions, MICAL1 is not an F-actin severing enzyme, as initially thought[23], and that its primary effect is to enhance filament depolymerization from both ends. We confirmed our observations using *Drosophila* MICAL (Supplementary Fig. 7c,d), and verified that MICAL is not altering actin dynamics through $H_2O_2$ production (Supplementary Fig. 7e), as previously reported[23]. Enhanced depolymerization is consistent with the notion that MICAL1 oxidation weakens longitudinal interactions between F-actin subunits. We expect this weakening to make filaments more sensitive to fragmentation-inducing conditions, and thus believe that the initially observed severing[23] was provoked by filament-surface interactions, different degree of bending, high fluorescent labelling fractions and/or illumination conditions. In fact, MICAL-oxidized filaments were recently reported to be more easily fragmented by mechanical stress and by the severing activity of cofilin[28]. This result also suggests that cofilin could contribute to removing F-actin from the abscission site in response to MICAL activation.

Its high depolymerizing activity implies that MICAL1 must be tightly regulated both in space and time. *In vitro,* the catalytic activity of MICAL1 monooxygenase domain[26,27] is modulated by its non-catalytic CH, LIM and C-terminal domains (refs 24,56 and see below). Importantly, it has been reported that MICAL and MICAL-like family members exist in an auto-inhibited, folded conformation[33,51]. Although MICAL-L1 lacks the mono-oxygenase domain, an intra-molecular interaction can be displaced when Rab13 is overexpressed in cells[57], but its functional relevance remains elusive. Previous studies reporting dMICAL roles in axon guidance in *Drosophila* revealed that the C-terminal part of the protein interacts with the semaphorin receptor PlexinA, which has been proposed to induce enzyme activity[29,51]. How MICAL is activated outside neurons is, however, poorly understood, and control of cytokinetic abscission by extracellular ligands such as semaphorin seems unlikely. We found using recombinant proteins that MICAL monooxygenase/FAD-CH-LIM domain directly interacts with the C-terminal half of the protein. The folded conformation of full-length MICAL1 displays low enzymatic activity in single actin filament assays presented here, bulk assays[22–24], or when overexpressed in interphase cells[33]. Remarkably, the presence of GTP-Rab35 is able to displace the intramolecular interaction and fully release inhibition of the enzyme. Altogether, we propose that MICAL1 binding to Rab35 not only localizes MICAL1 in late cytokinetic bridges, but also activates monooxygenase activity.

Oxidoreduction is one of the most fundamental processes in living organisms and plays a pivotal role in metabolic reactions. In a disease perspective, oxidative stress generates ROS that contributes to aging by oxidizing proteins, nucleic acids and lipids in a non-specific manner[58]. In contrast, this study highlights the critical role of controlled actin oxidation in cytoskeleton dynamics and reveals an unexpected role of oxidoreduction in cell division.

## Methods

**Cell cultures.** HeLa cells (ATCC)[59] were grown in DMEM medium (Gibco BRL) supplemented with 10% fetal bovine serum in 5% $CO_2$ condition at 37 °C. *Drosophila* S2 cells[39] were grown in Scheinder medium (Invitrogen) at 26 °C. Anillin-mCherry S2 cell line has been generated and characterized in ref. 40. For rescue experiments, HeLa cells were treated with 20 nM of Latrunculin-A (Sigma-Aldrich). For F-actin stabilization, HeLa cells were treated with 50 nM of Jasplakinolide (Santa Cruz Biotechnology) for 4 h before fixation.

**Genome-edited cell lines.** TALEN-edited HeLa cell lines for GFP-Rab35 have been generated and characterized in ref. 19. GFP-actin HeLa cells were designed by Cellectis Bioresearch SAS (Paris, France). GFP-MICAL1 HeLa cell line has been obtained by homologous recombination after cut by CRISPR/Cas9 at the locus. A plasmid encoding Cas9 enzyme (Addgene) with the following guide sequence: 5′-GGAGGTAGGTGAAGCCATGG-3′ was co-transfected in HeLa cells with a plasmid for recombination containing 728 bp of the genomic region upstream to MICAL1 start codon, followed by the complete cDNA encoding eGFP, and 784 bp of the genomic region downstream to MICAL1 start codon. GFP-positive cells were

sorted by FACS and isolated clones were analysed by genomic PCR and in western blot for GFP-MICAL1 recombinant protein expression using anti-MICAL1 antibodies.

**Plasmids and siRNAs.** Human MICAL1 has been amplified by PCR with reverse transcription (RT–PCR) from HeLa cells and introduced into pENTR gateway vectors, then recombined into pGFP and pCherry destination vectors. GFP-MICAL1$^{3G3W}$, GFP-MICAL1$^{1-1026}$, GFP-MICAL1$^{mutantS1}$, GFP-MICAL1$^{I1048R}$, GFP-MICAL1$^{R1055E}$ and GFP-MICAL1$^{M1015R}$ have been obtained using Quickchange (Agilent). pCMUIV empty (control), pCMUIV Rab35$^{S22N}$ and pGEX4T1 Rab35 have been described in ref. 43. pGAD and pLex vectors and constructs have been described in ref. 17. pmCherry human Rab35, pFlag empty (control), pFlag-Rab35$^{WT}$, pFlag-Rab35$^{Q67L}$, pFlag-Rab35$^{S22N}$ have been described in ref. 19. For rescue experiments, siRNA-resistant versions of GFP-MICAL1$^{WT}$, GFP-MICAL1$^{3G3W}$ and GFP-MICAL1$^{1-1026}$ have been obtained by mutating 6 bp of the siRNA-targeting sequence using Quickchange (Agilent). All point mutations have been generated using Quickchange (Agilent).

siRNAs against human *MICAL1*: 5′-GAGUCCACGUCUCCGAUUU-3′, and control siRNA-directed against *Luciferase*: 5′-CGUACGCGGAAUACUUCGA-3′ have been synthetized by Proligo-Sigma. RNAi in Drosophila S2 cells has been carried out as described in ref. 39, using the targeting sequence amplified by PCR using the following primers: Forward: 5′-ACTTTAGGAGGAAGGAGTTCCG-3′, Reverse: 5′-CACGGTATAGGCACTGATGTCC-3′. S2 cells were incubated for 6 days with dsRNAs and movies were recorded for an additional 2 days. Efficiency of RNAi was checked by RT–PCR using the following primers: Primer sequence for GAPDH: Forward: 5′-CGAATGTGGTTGCCGTGCC-3′, Reverse: 5′-GTGGTTCG CCTGGAAGAGA-3′. Primer sequence for *dMical*: Forward: 5′-CAGAGATCCG ATGATGAGAG3-3′, Reverse: 5′-CATCGCGTTTCTTGAGTGCT-3′.

**Antibodies.** The following antibodies were used for western blot procedures: mouse anti-β-tubulin (1:5,000, Sigma T5168), rabbit anti-MICAL1 (1:500, Proteintech Europe 14818-1-AP), rabbit anti-Rab35 (ref. 43), mouse anti-His (1:2,000, Sigma H1029), mouse anti-GST (1:2,000, BD Pharmingen 554805) and anti-Flag antibodies (1:1,000, Sigma M2 F1804). The following antibodies were used for immunofluorescence experiments: mouse anti-β-tubulin (1:200, DSHB E7), mouse anti-Aurora B (1:200, BD Bioscience 611082), rabbit anti-CHMP4B (1:200, Santa Cruz Biotechnology 82557), rabbit anti-p34-Arc/ARPC2 (1:200, Millipore 07-227). The following secondary antibodies were used: Dylight Alexa 488- and Cy3- and Cy5-conjugated secondary antibodies (Jackson Laboratories) were diluted 1:500.

**Cell transfection.** Plasmids were transfected in HeLa cells for 24 or 48 h using X-tremeGENE 9 DNA reagent (Roche). For MICAL1 silencing experiments, HeLa cells were transfected twice with 50 nM siRNAs for 96 h using HiPerFect (Qiagen) following the manufacturer's instructions. In cases of rescue experiments, cells were first transfected for 72 h with siRNAs using HiPerFect, then by plasmids using CaCl$_2$ precipitates for an additional 24 h.

**Western blot.** Western blot experiments after siRNA treatment were carried out as follows[60]:, cells treated with siRNAs were lysed in NP-40 extract buffer (50 mM Tris, pH 8, 150 mM NaCl, 1% NP-40) containing protease inhibitors. Ten microgram of lysate were migrated in 12% SDS–PAGE (Bio-Rab Laboratories), transferred onto PVDF membranes (Millipore) and incubated with corresponding antibodies in 5% milk in 50 mM Tris-HCl pH 8.0, 150 mM NaCl, 0.1% Tween20. followed by HRP-coupled secondary antibodies (1:20,000, Jackson Immuno Research) and revealed by chemiluminescence (GE Healthcare). Uncropped western blots are presented in Supplementary Fig. 9.

**Immunofluorescence and image acquisition.** Methanol fixation (3 min at −20 °C) has been used for CHMP4B staining. For all other antibodies, HeLa cells were grown on coverslips and fixed with PFA, one volume of 8% paraformaldehyde (PFA) directly added in the culture medium (1:1 volume) for 10 min at room temperature and then replaced by 4% PFA for 10 min. Cells were then processed for immunofluorescence. Cells fixed in PFA were permeabilized and blocked with PBS containing 0.2% BSA and 0.1% saponin and successively incubated for 45 min at room temperature with primary and secondary antibodies diluted in PBS containing 0.2% BSA and 0.1% saponin[41]. Cells fixed with methanol were blocked with PBS containing 0.2% BSA and successively incubated for 45 min at room temperature with primary and secondary antibodies diluted in PBS containing 0.2% BSA. Cells were mounted in Mowiol (Calbiochem). Phalloidin staining (1:2,000). DAPI staining (0.5 mg ml$^{-1}$, Serva). Images were acquired with an inverted Ti E Nikon microscope, using a × 100 1.4 NA PL-APO objective lens or a × 60 1.4 NA PL-APO VC objective lens and MetaMorph software (MDS) driving a CCD camera (Photometrics Coolsnap HQ). Images were then converted into 8- bit images using ImageJ software (NIH). Images in Figs 1c,d, 3b and 4e were acquired using an inverted Eclipse Ti E Nikon microscope equipped with a CSU-X1 spinning disk confocal scanning unit (MDS), driving a EMCCD Camera (Evolve

512 Delta, Photometrics). Images were acquired with a × 100 1.4 NA PL-APO objective lens or × 60 1.4 NA PL-APO VC and MetaMorph software (MDS).

**Time-lapse microscopy.** For time-lapse phase-contrast and fluorescent microscopy, HeLa cells were plated on 35 mm glass dishes (MatTek) and put in an open chamber (Life Imaging) equilibrated in 5% CO$_2$ and maintained at 37 °C. Time-lapse sequences were recorded every 5 or 10 min for 24 or 72 h using a Nikon Eclipse Ti (more details in section Immunofluorescence) inverted microscope with a × 20 0.45NAPlan FluorELWD or × 60 1.4 NA PL-APO VC objective controlled by Metamorph software (Universal Imaging).

**Correlative light and SEM.** HeLa cells that express GFP-Actin were used to select intercellular bridges of appropriate stage using fluorescent light microscope, and subsequently to re-localize the same cells in SEM Method for SEM preparation is described in details in ref. 42. Briefly, cells were successively fixed with 2.5% glutaraldehyde (Sigma-Aldrich), postfix with 2% osmium tetroxide (Electron Microscopy Science), dehydrated into baths of ethanol and dried into the critical point dryer's chamber (Leica EM CPD300). Samples were coating with 8 nm of gold/palladium (Gatan Model 681) and analysed with the JEOL6700 microscope for SEM acquisition.

**Recombinant protein purification.** GST-Rab35 and 6xHis-tagged MICAL (full-length or truncated versions) have been induced in *E. coli* and purified by affinity chromatography. Rab35 has been exchanged with either GDP or GTPγS using EDTA treatment.

GST-Rab35$^{WT}$ (encoded by pGEX-4T1-Rab35$^{WT}$) and GST alone (pGEX4T1 empty) were expressed in the BL21 pLysS strain of *Escherichia coli* after induction with 1 mM isopropyl-β-D-thiogalactopyranoside at 37 °C for 3 h. Cells were lysed in PBS containing 1 mg ml$^{-1}$ Lysozyme, 1 mM dithiothreitol and protease inhibitors (Roche) by sonication on ice.

6xHis-fused proteins drosophila MICAL$^{1-669}$, and human MICAL1$^{WT}$, MICAL1$^{FAD}$, MICAL1$^{FAD-CH-LIM}$, MICAL1$^{879-1067}$, MICAL1$^{879-1026}$ and GST-MICAL1$^{879-1067}$ were expressed in the BL21-AI (ThermoFisher scientific) strain of *Escherichia coli* after induction with 1 mM isopropyl-β-D-thiogalactopyranoside at 16 °C for 24 h (Thermo Fisher Scinetific). 6 × His-fused proteins were affinity-purified using Ni-NTA Magnetic Agarose Beads (Qiagen) and were eluted in 50 mM Tris pH 8, 150 mM NaCl, 2 mM MgCl$_2$ and 250 mM Imidazole. The GST fusion proteins were affinity-purified using glutathione Sepharose 4B (GE Healthcare) and eluted with 20 mM HEPES at pH 7.5, 150 mM NaCl and 20 mM reduced glutathione. All MICAL1 purified proteins were dialyzed at 4° overnight in 50 mM Tris pH 8, 150 mM NaCl, 1 mM DTT.

**GST-pulldown assay.** For direct binding assays, GST-Rab35$^{WT}$ was exchanged with either 1 mM GDP or 200 µM GTPγS in 25 mM Tris pH 7.5, 100 mM NaCl, 10 mM EDTA, 5 mM MgCl$_2$ and 1 mM DTT for 1 h at 37 °C. Nucleotides were then stabilized with 20 mM MgCl$_2$. GST proteins were loaded onto glutathione Sepharose 4B beads (Pharmacia) in 25 mM Tris pH 7.5, 1 mM MgCl$_2$ and 0.2% BSA for 1 h at 4 °C. Beads were then incubated with 6xHis-MICAL1 proteins in 50 mM Tris pH 8, 150 mM NaCl, 2 mM MgCl$_2$, 1 mM DTT. Beads were washed three times, resuspended into 1 × Laemmli buffer and boiled at 95 °C for 10 min. Pulled-down GST-Rab35 proteins loaded on beads were detected by Ponceau red staining and 6xHis-tagged proteins were detected by western blot using anti-6xHis antibodies (1:5,000).

For direct binding and competition assay, GST-MICAL1$^{879-1067}$ or GST alone were loaded onto glutathione Sepharose 4B beads (Pharmacia) in 25 mM Tris pH 7.5, 1 mM MgCl$_2$ and 0.2% BSA for 1 h at 4 °C. Beads were then incubated with 6xHis-MICAL1$^{FAD-CH-LIM}$ alone or with increasing amount of Rab35-GppNHp proteins in 50 mM Tris pH 8, 150 mM NaCl, 2 mM MgCl$_2$, 1 mM DTT. Beads were washed three times, resuspended into 1 × Laemmli buffer and boiled at 95 °C for 10 min. Pulled-down GST-Rab proteins loaded on beads were detected by Ponceau red staining, 6xHis-tagged proteins were detected by western blot using anti-6xHis antibodies (1:5,000) and Rab35 proteins were detected by western blot using anti-Rab35 antibodies (1:500).

**Co-immunoprecipitation experiments.** HEK 293T cells were transfected with Flag-tagged control (empty) or Rab35 constructs (Rab35$^{WT}$, Rab35$^{S22N}$ or Rab35$^{Q67L}$) for 36 h. Cells were lysed in a buffer containing 20 mM Tris at pH 7.4, 100 mM KCl, 2 mM MgCl$_2$, 10% glycerol and 0.1% NP-040 and phosphatase and protease-inhibitors. Post-nuclear supernatants were incubated with Protein G Sepharose Beads (Protein G Sepharose, GE HealthCare) for 30 min (preclarification). Supernatants were then incubated with M2-Flag agarose (Sigma) 1 h 30 min at 4 °C. After three washes in lysis buffer, proteins were resuspended in Laemmli buffer and boiled for 10 min at 95 °C. The amount of co-immunoprecipitated MICAL1 in each condition was probed by western blotting using the indicated antibodies.

**Yeast two-hybrid experiments.** A yeast two-hybrid experiments were performed by co-transforming the *Saccharomyces cerevisiae* reporter strain L40 with either pGAD-MICAL1[879-1026], pGAD-MICAL1[879-1067], pGAD-MICAL1[918-1067], as well as, MICAL1[879-1067] mutants S1 (E946K, V950D, E953K, V971E, L975R, E978R), S2 (R1012E, M1015R, L1034K, V1038E), S3 (E1101R, V1038E, V1041E, I1048R, R1055E) and MICAL1[879-1067] single mutants M1015R, I1048R and R1055E, together with either pLex-human Rab35[WT], pLex-human Rab35[Q67L], pLex-human Rab35[S22N]. Transformed yeast colonies were selected on DOB agarose plates without Tryptophane and Leucine. Colonies were picked and grown on DOB agar plates with Histidine to select co-transformants and without Histidine to detect interactions.

**Statistical analysis.** All values are displayed as mean ± s.d. for at least three independent experiments (as indicated). Significance was calculated using unpaired $t$-tests, $\chi^2$-tests or two-way analysis of variance (ANOVA) tests, as indicated. For abscission times, a non-parametric Kolmogorov–Smirnov test was used. In all statistical tests $P > 0.05$ was considered as not significant. By convention, $*P < 0.05$; $**P < 0.01$ and $***P < 0.001$.

**Single filament experiments.** Actin was purified from rabbit muscle acetone powder and labelled with Alexa 488. In microfluidics experiments, filaments were aged for 15 min with $0.12\,\mu M$ actin to become fully ADP-actin, prior to depolymerization[50]. In surface-anchored experiments, steady-state filaments were anchored and exposed to depolymerization conditions $\sim 120\,s$ before observation[23]. Images were acquired in epifluorescence or TIRF microscopy, and were analysed using Image J and homemade Python software.

*Proteins and buffers.* Actin was purified from rabbit muscle acetone powder (Pel-freez) using the following protocol[61]: the powder was resuspended in G-buffer (5 mM TRIS pH 7.8, 0.2 mM ATP, 0.1 mM CaCl$_2$, 1 mM DTT and 0.01% NaN$_3$) and centrifuged at 18,000 r.p.m., keeping the supernatant which was then filtered through cheese cloth. To remove contaminants, 3.3 M KCl was added and the solution was centrifuged at 18,000 r.p.m., keeping the supernatant, which was again filtered through cheese cloth. The solution was dialysed overnight in 1 mM MgCl$_2$ and 100 mM KCl to polymerize filaments, followed by the addition of 800 mM KCl to dissociate contaminating proteins from the filaments. After centrifugation at 35,000 r.p.m., the pellet was resuspended and dialyzed in G-buffer to depolymerize. Aggregates were removed from the solution by centrifugation at 35,000 r.p.m. Remaining contaminants were eliminated by gel filtration in a Superdex 200 column (GE Healthcare). G-actin was stored in G-buffer: 5 mM TRIS pH 7.8, 0.2 mM ATP, 0.1 mM CaCl$_2$, 1 mM DTT and 0.01% NaN$_3$. Filament elongation and depolymerization were carried out in standard F-buffer: 5 mM TRIS pH 7.8, 0.2 mM ATP, 50 mM KCl, 1 mM MgCl$_2$, 0.2 mM EGTA, 10 mM DTT and 1 mM DABCO. We used $40\,\mu M$ of H$_2$O$_2$ (Sigma-Aldrich) for Supplementary Fig. 7e. Labelling was done by incubating F-actin with Alexa488 succimidyl ester (Molecular probes), thereby labelling Lysines on the outer surface of polymerized actin. Experiments were carried out with 15% labelLed actin. Labelling had no incidence on the measured actin assembly rates.

*Single filament experiments with microfluidics.* Microfluidics experiments (Fig. 6a–c; Supplementary Fig. 7c–e) were done in PDMS microchambers with 3 or 4 entry channels and mounted on a clean glass coverslip[50]. Typical dimensions of the microchamber were as follows: 20 to 100 micrometres high, 1 mm wide, and 1.5 cm long. Flow rates ranged from 300 to 10 000 nl min$^{-1}$. Control experiments were carried out to verify that chamber dimensions and flow rates had no incidence on the measured actin assembly rates. Spectrin-actin seeds were adsorbed on the coverslip surface, followed by Bovine Serum Albumin (BSA, Sigma) for passivation. A solution of $1\,\mu M$ G-actin was flown in to elongate filaments, for a period of 5 to 15 min. Filaments were then exposed to a critical concentration of $0.12\,\mu M$ G-actin for 15 min, allowing them to fully hydrolyse ATP and release inorganic Phosphate while keeping a constant length. The resulting ADP-actin filaments were then exposed to buffer or protein solutions, and their depolymerization was monitored. In the microfluidics set-up, the solution to which the filaments are exposed is changed in less than a second.

In the presence of FAD + NADPH, filaments typically depolymerized in <200 s. However, due to photoinduced pauses, which are an independent phenomenon that we have characterized elsewhere[62], some filaments resumed depolymerization after a pause of a few minutes, thereby allowing us to measure the depolymerization rate over longer time scales, as shown in Fig. 6b.

In the kymograph on the left of Fig. 6c, depolymerization conditions were changed from FAD + NADPH to buffer alone, 85 s after the onset of depolymerization. In the kymograph on the right of Fig. 6c, long ADP-actin filaments were exposed to FAD + NADPH for 150 s, before being elongated again from fresh G-actin in F-buffer (no FAD), then aged again for 15 min, and finally depolymerized in buffer.

*Single filament experiments with multiple surface anchoring.* Surface-anchored experiments (Fig. 6d,f; Supplementary Fig. 7a) were done as follows[23]: flow channels were built with a clean coverslip mounted on a microscope slide, using parafilm as a spacer. Each channel was incubated with inactivated myosin, obtained by incubating rabbit muscle myosin (Cytoskeleton Inc.) with *N*-ethylmaleimide (Sigma). Channels were then passivated with BSA and rinsed, before diluting and flowing in a solution of steady-state actin filaments. Unbound filaments were

washed out with buffer, and the depolymerizing solution was flown in. The delay between the introduction of the depolymerizing solution in the sample and the start of the microscope acquisition was typically of 1.5 to 2 min (unlike microfluidics experiments, where depolymerization was monitored from time zero).

In these experiments, depolymerizing filaments sometimes exhibited irregular kymographs, with multiple short pauses, in addition to the photoinduced pauses mentioned above[62]. These are certainly due to the multiple interactions between the surface myosins and the filaments, as already reported in the first experiments of this type[63]. These events were excluded from our analysis, and we only fitted periods of clear depolymerization occurring steadily over several consecutive time frames.

**Image acquisition and analysis.** Images were acquired with $\times 60$ magnification, on either a Nikon Te2000 inverted microscope in epifluorescence with an Xcite 120Q light source (Lumen Dynamics) and an Orca-flash2.8 camera (Hamamatsu); or on a Nikon Ti-E Eclipse inverted microscope, either in epifluorescence with an Xcite exacte light source (Lumen Dynamics) and an Orca-flash4.0 camera (Hamamatsu), or in TIRF with an ILAS2 (Roper), a 150 mW 488 nm laser and an Evolve 512 EMCCD camera (Photometrics). Epifluorescence images were acquired using μ-manager, TIRF acquisitions were carried out using MetaMorph. During filament depolymerization, images were typically acquired every 5 or 10 s.

Images were analysed with image J or using homemade software written in Python. In Image J, the Multiplekymograph plugin was used to generate the kymographs shown in Fig. 6 and Supplementary Fig. 7. The depolymerization rates shown in Fig. 6f and Supplementary Fig. 7a were determined by measuring the slope of regular portions of these kymographs, excluding pausing events, be they photoinduced or caused by multiple surface anchors. The depolymerization rates of Fig. 6b and Supplementary Fig. 7d,e were determined with the Python software: filament lengths were determined from intensity linescans of their contour, and tracked over time. Depolymerization rates were determined by a linear fit of the length over 3 consecutive time points (that is, a total interval of 20 s). Photoinduced pauses were excluded from the analysis.

**Recombinant protein production for structural studies.** Recombinant expression of Rab35[1-175], Rab35[1-199], MICAL1[879-1067], MICAL1[918-1026] and MICAL1[918-1067] WT (as well as, MICAL1[918-1067] mutants S1 (E946K, V950D, E953K, V971E, L975R and E978R), S2 (R1012E, M1015R, L1034K and V1038E), S3 (E1001R, V1038E, V1041 E, I1048R and R1055E) and single mutants M1015R, I1048R and R1055E) were performed in *Escherichia coli* BL21-CodonPlus-RILP or BL21-Gold cells using a pPROEX-HTb vectors containing an N-terminal 6xHis peptide and rTEV cleavage site.

Bacterial cells were grown at 37 °C in LB medium, induced at an $A$ 600 nm of 0.6 by the addition of 1 mM isopropyl-1-thio-β-D-galactopyranoside, and collected after 18 h at 20 °C. The cell pellet was resuspended in 50 mM Tris pH 8.0, 300 mM NaCl, 2 mM MgCl$_2$, 30 mM imidazole, 2 mM 2-mercaptoethanol, 5% glycerol and protease inhibitor mix (Chymostatin, Leupeptin, Antipain, Pepstatin A at $1\,\mu g\,ml^{-1}$), lysed by sonication, and centrifuged at 35,000 $g$ for 1 h. The supernatant was loaded onto a HisTrap column (GE Healthcare). After washing with the lysis buffer, the fusion proteins were eluted with 300 mM imidazole added in the same buffer. The his-tag was cleaved by incubation with rTEV protease at 1:50 molar ratio overnight. The protein was further purified by gel filtration on Superdex-75 (GE Healthcare) in 50 mM Tris pH 8.0, 150 mM NaCl, 2 mM MgCl$_2$, 2 mM TCEP, 5% glycerol, concentrated using Vivaspin turbo (Sartorius) and stored at −80 °C. For the production of selenomethionine derivatized MICAL1[918-1067] we used the methionine auxotroph *E. coli* strain B834(DE3) and SelenoMet Medium (Molecular Dimensions).

For nucleotide exchange, Rab35 was incubated with a 20-fold molar excess of GppNHp, in 50 mM Tris pH 8.0, 150 mM NaCl, 2 mM TCEP, 5% glycerol and 5 mM EDTA. The exchange reaction was quenched by addition of 10 mM MgCl$_2$, and excess nucleotide was removed by gel filtration chromatography.

**Crystallization and structure determination.** The crystallization experiments were performed at 17 °C by vapour diffusion in hanging drops. Crystals of the Se-Met MICAL1[918-1067] were grown in 35% ethylene glycol and 4% 1,6-hexanediol. The crystals were flash-frozen in liquid nitrogen. An anomalous X-ray diffraction data set was collected to 3.3 Å resolution at Soleil synchrotron PX1 beamline. The X-ray diffraction data were indexed, integrated and scaled with the XDS program suite[64] The structure was determined by single-wavelength anomalous diffraction method with the Phenix (Autosol)[65]. The program determined positions of two selenium atoms (see the anomalous difference map in Supplementary Fig. 3a), subsequent phasing and density modification resulted in an interpretable electron density map allowing to build an initial model with Phenix (Autobuild)[65]. The model was refined using the Phenix (Refine) programs[65] and manual rebuilding using COOT[66]. The data collection and refinement statistics are summarized in Supplementary Table 2.

**Isothermal titration calorimetry.** Isothermal titration calorimetry was performed using a MicroCal ITC-200 titration microcalorimeter (Malvern) at 10 °C. The

C-terminal MICAL1 domain at a concentration of 60 μM in the interaction buffer (50 mM Tris buffer, pH 8, 150 mM NaCl, 2 mM MgCl$_2$, 2 mM TCEP and 5% Glycerol), was placed in the calorimeter cell, and GppNHp-Rab35 (480 μM in the interaction buffer) was added sequentially in 2 μl aliquots. The heat of reaction per injection (microcalories per second) was determined by integration of the peak areas using the Origin software and provided the heat of binding, the binding stoichiometry, and the interaction association constant using a one-site model. The heats of dilution were determined in parallel control experiments by injecting GppNHp-Rab35 (480 μM in the interaction buffer) into the interaction buffer. The heat of dilution was subtracted from the interaction heat before fitting the curve.

**SEC-MALS.** Absolute molar masses of proteins were determined using size-exclusion chromatography combined with multi-angle light scattering (SEC–MALS). Protein samples (50 μl; 10 mg ml$^{-1}$) were loaded onto a Superdex-75 column (GE Healthcare) or XBridge BEH SEC 200 Å in 50 mM Tris pH 8.0, 150 mM NaCl, 2 mM MgCl$_2$, 2 mM TCEP, 5% glycerol, at 0.5 ml min$^{-1}$ using a Dionex UltiMate 3000 HPLC system. The column output was fed into a DAWN HELEOS II MALS detector (Wyatt Technology). Data were collected and analysed using Astra X software (Wyatt Technology). Molecular masses were calculated across eluted protein peaks.

**SAXS measurements and data analysis.** Small angle X-ray scattering (SAXS) data were collected on the SWING beamline (synchrotron SOLEIL, France). MICAL1$^{918-1067}$ WT, S1, S2 and S3 constructs 50 μl at 10 mg ml$^{-1}$ were injected on the online HPLC system (Agilent SEC-5 500 Å) in 50 mM Tris pH 8, 150 mM NaCl, 2 mM TCEP, 2 mM MgCl$_2$, 5% glycerol. Twenty to 29 frames of 1.5 s exposure (and 45 for the buffer) were averaged and buffer scattering was subtracted from the sample data. The radius of gyration and the intensity at the origin were determined using the Guinier law[67]. Side chains and loops missing in the X-ray structure were modelled using *COOT*[66]. FoXS web-server[68] was used to fit the theoretical scattering intensity from the X-ray structure into the experimental SAXS data.

**Data availibility.** The authors declare that the main data supporting the findings of this study are available within the article and its Supplementary Information files. The atomic coordinates and structure factors of the MICAL1 C-terminal domain structure has been deposited in the Protein Data Bank (www.pdb.org) with accession number PDB: 5LE0 (Supplementary Table 2).

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

## Acknowledgements

We thank R. Basto, A. Gautreau and G. Langsley for critical reading of the manuscript; the Echard Lab members for helpful discussions; the Romet/Jegou Lab members for help with in vitro experiments; G. Hickson, the Plateforme Anticorps Recombinants (Institut Curie, Paris) and the DHSB (University of Iowa) for reagents and plasmids. We thank the imaging facilities Imagopole and Ultrapole Institut Pasteur. We thank P.H. Commere from the Cytometry platform Institut Pasteur for FACS sorting. We thank beamline scientists of PX1 (SOLEIL synchrotron) for excellent support during data collection. This work has been supported by Institut Pasteur, CNRS, FRM (Equipe FRM DEQ20120323707), INCa (2014-1-PL BIO-04-IP1), ANR (AbCyStem) and IXCORE foundation to A.E., J.B. and K.K. have been awarded a doctoral fellowship from the Pasteur Paris Universités International PhD program and Carnot-Pasteur MI, and a fellowship from FRM (FDT20150532389). A.H. was supported by grants from the CNRS and INCa (2014-1-PL BIO-04-ICR-1). H.H. has been awarded a doctoral fellowship from the PSL Université. The A.H. team is part of Labex CelTisPhyBio 11-LBX-0038, which is part of the Initiative d'Excellence at PSL Research University (ANR-10-IDEX-0001-02 PSL). H.W. has been supported by a postdoctoral grant from the Fondation ARC pour la recherche sur le cancer.

## Author contributions

S.F. carried out the experiments presented in Figs 1,2,3,4a–d,4f–i,5d,6e, Supplementary Figs 1a–d,f,g; Supplementary Figs 6,8; K.K. and J.B. the experiments in Fig. 4e and Supplementary Fig. 1e, respectively; G.R.-L. the experiments in Fig. 6a–d,f and Supplementary Fig. 7. G.R.-L. and H.W. quantified single filament experiments; H.W. developed software for analysis. H.H. and O.P. determined the X-ray structure and analysed it with A.H. MALS and SAXS experiments were performed and analysed by H.H. and C.K., with the help of V. Ropars. ITC data were collected by H.H. and D.S. A.E. conceived the project, A.E., A.H. and G.R.-L. oversaw the experiments. S.F., H.H., G.R.-L., A.H. and A.E. designed and interpreted the experiments. A.E., S.F., O.P., A.H. and G.R.-L. wrote the manuscript with help from O.P. A.E. and A.H. secured funding.

## Additional information

**Competing financial interests:** The authors declare no competing financial interests.

