## [Peer Review File · Nature Communications]

RESPONSE TO REFEREES

We thank the three Reviewers for their very positive comments and for suggested experiments. We have provided additional experiments and Figures as well as explanations in the revised manuscript to address all their comments. In the revised manuscript, we have reorganized and shortened the introduction and the discussion in order to comply with *Nature Communications* guidelines.

New experimental data have been added as. 3e, 3f, 4d, Supplementary Fig. S1f, S1g, S2j, S6d, S7c, S7d, S7e, S8a and S8b.

We provide below a full response to each comment raised by the Reviewers.

We believe that the added experimental data and Reviewer's suggestions helped us to reinforce the conclusions of the manuscript.

Reviewer #1 (Remarks to the Author):

Summary: This manuscript presents data that explores the role of MICAL1 in cell abscission, the final step of cell division. MICAL1 is an NADPH oxidase that targets actin oxidation and one hypothesis being tested in this manuscript is that MICAL1 is required to remove actin in the final stages of abscission. The mechanism by which MICAL1 gets localized to the bridge region of the abscission zone and the mechanism by which MICAL1 depolymerizes the F-actin in this region are also addressed, as is the relationship between the localization of MICAL1 and CHMP4B, a component of the ESCRT complex previously shown to co-localize with spastin, a microtubule severing protein needed to clear microtubules from the bridge region before abscission. It was previously known that MICAL1 interacted with Rab35 and herein the authors show this interaction is essential for MICAL1 localization to the cytokinetic bridge Furthermore they identify the specific region of MICAL1 involved in Rab35 binding, discovering a novel three helix interaction in the process. They show that silencing MICAL1 delays the average time to abscission by quite a bit, although about 40-50% of cells do seem to undergo abscission in the normal time frame, suggesting that there are probably some redundant mechanisms at play in this process, a point not discussed in the manuscript even though additional evidence suggesting this is presented. The most surprising result of the manuscript is the finding that MICAL1 has no F-actin severing activity, as has been reported for *Drosophila* MICAL, but rather that its activity increases the rate of filament depolymerization from both ends through an NADPH dependent step, presumably the oxidation of the met44 of actin. This was shown most dramatically by an excellent experiment in which fluorescently tagged oxidized actin was elongated with fluorescently tagged normal actin and its rate of depolymerization followed in the absence of MICAL1. The filaments depolymerized slowly until the zone of oxidized subunits was reached and then rapid depolymerization ensued. The authors go to great lengths to show that their results hold up under their favored depolymerization conditions (F-actin that is end anchored) as well as the identical conditions used by others to show severing (linked by a dead myosin II to the surface).

Critique: This is an excellent paper that offers insight into one mechanism of actin removal from the cytokinetic bridge preceding abscission. The manuscript is well written, the science is sound, the methods are appropriate and the results are convincing.

We thank the Reviewer for this very positive evaluation of our work.

However, there are multiple ways for abscission to occur and the authors own evidence clearly supports this assumption:

1. Only 62% of cytokinetic bridges containing the ESCRT complex component CHMP4B have detectable pools of MICAL
2. Depletion of MICAL1 only entirely blocked abscission in 4% of cells (3 fold more than the control cell population, but not terribly impressive if MICAL1-dependent actin disassembly was the only mechanism for the actin removal to occur).
3. Although the average abscission time increased in MICAL1 silenced cells, about 40-50% of the cells completed abscission at nearly identical times as the wild type (control) population. Thus a bit more discussion on cooperative or redundant mechanisms would be appropriate.

We now provide a revised discussion specifically addressing this point (p. 20). In particular, we mention that additional pathways such as Rab35/OCRL and Rab11/p50RhoGAP (or other pathways) likely cooperate to regulate such an important step in cytokinesis.

In dividing cells, cofilin is also enriched within the cytokinetic bridges and may provide an alternative (or indeed even a cooperative) mechanism for actin removal. Work recently published by Grintsevich et al (NCB July 2016) demonstrates MICAL oxidized actin actually recruits cofilin to enhance the rapid depolymerization of the actin. Although MICAL-induced severing is also reported in the Grintsevich et al paper, the rapid loss of actin following cofilin addition to MICAL oxidized actin occurred in the absence of MICAL, showing that the oxidation lead to cofilin-enhanced depolymerization. While it is not incumbent upon these authors to provide a complete analysis of cofilin in their model, acknowledgement of other mechanisms, either redundant or cooperative, should be mentioned in the discussion. This is especially pertinent because the LatA rescue of abscission in MICAL depleted cells shown in this paper should only occur if the F-actin population is dynamic and this dynamics is usually dependent upon the presence of cofilin along with its entourage of proteins (coronin, Aip1, etc) that help in its recruitment and severing/depolymerizing functions.

Cofilin has a crucial role in furrow ingression by regulating actin ring dynamics. To our knowledge, the localization of cofilin later in cytokinetic bridges has not been reported. Our unpublished data indicate that a pool of cofilin is indeed detected in bridges, suggesting that MICAL1 and cofilin might cooperate during abscission. Since cofilin has a crucial role during furrow ingression, it is however technically challenging to test this hypothesis experimentally. We nevertheless discuss this interesting possibility in the light of the Grintsevich et al. paper on p. 21, as suggested.

With regard to newly proposed mechanism of F-actin depolymerization from either end by MICAL without severing (which I think is very clear and nicely demonstrated here) it should be noted that the previously published papers showing the severing mechanism utilized *Drosophila* MICAL and not mammalian MICAL1 used here. The authors' suggest that severing observed in these other assays may be artefactual due to photocleavage of oxidized fluorescent actin. While this may be the explanation, it is also possible that *Drosophila* MICAL has severing activity not found in mammalian MICALs. Thus the authors should obtain an aliquot of the *Drosophila* MICAL for a direct comparison. Otherwise, readers will be left to wonder if the other published studies are incorrect or if there are real differences between MICAL activities from different sources. This study is critical for publication of the paper in NCB because it directly contradicts some aspects of other work recently published. Indeed if the *Drosophila* MICAL does not sever at all, some results of the previous papers would need to be reinterpreted.

To extend our conclusions, we have confirmed our data using *Drosophila* MICAL in our own microfluidic set up and time scale, as requested. We used the same construct as in Hung et al. Science 2011 (dMICAL aa 1-669) and did not observe severing activity (new Fig. S7c-d). However, depolymerization from both ends was observed, as seen in our previous experiment using human MICAL FAD. We note that the latest report from the same group (Grintsevich et al. NCB 2016, which was published while our manuscript was being reviewed) now shows that severing of MICAL-oxidized filaments only occurs when filaments are stressed (pipetting) or exposed to cofilin. Altogether, we conclude that MICAL-dependent oxidation of F-actin weakens the filaments and that the primary consequence of this weakening is an enhanced depolymerization, not severing (discussion p. 21).

Minor corrections:

Page 4, line 2 and 3: end the sentence after microtubules- the first word on line 2- the remainder is redundant.

Page 4, line 7: The recruitment of the phosphatase to deplete PIP2 is clear, but the cause and effect conclusion of "therefore prevents F-actin accumulation in late..." is not so clear. It sounds like the authors are saying that F-actin cannot assemble in the absence of PIP2 which I am sure is not what the authors intended to imply. The wording here should be revised.

Page 8, last line and top of page 9: phalloidin is certainly used as a marker for F-actin but phalloidin does not bind F-actin saturated with cofilin- and if cofilin is preferentially recruited to MICAL oxidized actin, as work from Grintsevich et al implies, the loss of MICAL might reduce the cofilin recruitment which might make the filaments easier to label with phalloidin, accounting for some of the increase observed. Do the authors know if F-actin oxidation by MICAL affects phalloidin binding? Are the increased numbers of membrane bulges and extensions perhaps due to recruitment of Arp2/3 or formins to the midbody in the absence of MICAL oxidation of F-actin? Certainly the LatA effects described on page 9 indicate a dynamic actin pool in this region.

Page 10, last line: suggested revision ... at the abscission site was partially, but significantly, rescued by

Page 12, last paragraph, second sentence: suggested change: Whereas the C-terminal... with Rab35Q67L, removing the last...

Page 17, line 12: I believe 6e should be the referenced figure, not 6a as listed.

We modify the text, as requested.

In addition, we now provide evidence that indeed the p34^{Arc} subunit of Arp2/3 is clearly detected in MICAL1-depleted bridges (new Fig. S8b and discussion p. 20), confirming the presence of a dynamic pool of actin in bridges in this condition.

In addition, to rule out that potential interactions with cofilin would alter the original quantifications, we independently measured an increase of GFP-actin in bridges (using a genome-edited cell line expressing endogenous levels) in cytokinetic bridges after MICAL1 depletion (new Fig. S1f). Quantifications using labelled phalloidin and GFP-actin gave very similar results.

Reviewer #2 (Remarks to the Author):

In this manuscript Fremont and colleagues report that oxidation of F-actin controls cytokinetic abscission. The authors find that MICAL1, a flavoprotein monooxygenase known to oxidize F-actin on Met-44, is recruited to the midbody and abscission zone by the small GTPase Rab35. Interference with MICAL1 or its catalytic activity causes cytokinetic abscission delay, accompanied by increased actin levels in the cytokinetic bridge. Recruitment of ESCRT-III, thought to mediate membrane scission, is impaired under these conditions, and recruitment can be restored with the actin poison Latrunculin A. Using a microfluidics assay the authors find that MICAL1 induces actin depolymerization from both ends and does not function as an actin-severing enzyme as previously proposed. They solve the crystal structure of the Rab35-binding region of MICAL1 at 3.3 Å and show that it consists of a curved sheet of three helices. Evidence is presented that MICAL1 exists in an autoinhibited conformation which is relieved upon Rab35 binding.

The technical standard of this manuscript is excellent, and most of the conclusions are well supported by the data. Collectively these findings provide substantial new insight into the regulation of actin dynamics during cytokinetic abscission and should be of wide interest.

We thank the Reviewer for this very positive evaluation of our work.

Specific comments:

1. The structure of the manuscript is a bit strange. The authors start with localization and functional studies of MICAL1, and later they identify – seemingly out of nowhere – MICAL1 as an interactor of Rab35 in a yeast two-hybrid screen for Rab35 effectors. Surely this study must have started with the two-hybrid screen?

The Reviewer is right, but we wanted to centre the paper on the main finding: the role of MICAL1 in F-actin oxidation during cell division. Then, to unravel how Rab proteins recruit MICAL1 and activate the redox activity. To address the Reviewer's comment, we have now cited what was known about MICAL1/Rabs interaction in the literature in the introduction

(p. 4). We then characterize in details in the second part of the manuscript the relevance of the Rab35/MICAL1 interaction that we now demonstrate to be direct.

2. The weakest conclusion of the manuscript is the notion that MICAL1-mediated actin depolymerization is required for ESCRT-III recruitment. It is interesting that CHMP4B was correctly recruited in only 25% of the cytokinetic bridges in MICAL1 depleted cells (Fig 3e), but even in control cells only a minority of the cytokinetic bridges showed recruitment of CHMP4B. How can this be explained? And how would F-actin prevent ESCRT-III recruitment? Does stabilization of F-actin with Jasplakinolide prevent ESCRT-III recruitment?

We have rewritten the corresponding text to make it clearer, as these figures actually referred to CHMP4B at the abscission (secondary) site specifically and not at the whole bridge, which should reassure the Reviewer. Actually, 85% of the control bridges contain CHMP4B (either at the midbody or at the abscission site), i.e. in the great majority of the bridges, as expected (p. 10 and Fig. 3e-f).

In order to gain insights into the role of MICAL1-mediated actin depolymerization in ESCRT-III recruitment, we now provide additional quantifications of ESCRT-III recruitment at the midbody only vs. midbody + abscission site, both in control- and in MICAL1-depleted cells (new Fig. 3e and 3f). CHMP4B was correctly recruited at the abscission site in only 23% of the bridges in MICAL1-depleted cells, in contrast to 39% in control cells (Figure 3e). Concomitantly, an increase of bridges with CHMP4B localized only at the midbody was observed after MICAL1 depletion (Figure 3e). Altogether, these results suggest that clearing F-actin from the bridge does not perturb initial ESCRT-III recruitment at the midbody but its subsequent recruitment at the abscission site (p. 10).

Finally, we experimentally tested the effect of Jasplakinolide on ESCRT-III recruitment, as requested. Interestingly, Jasplakinolide treatment reduced ESCRT-III recruitment to the abscission site (new Fig. S8a and discussion p.20), suggesting that chemical stabilization of actin also perturbs the recruitment of the abscission machinery.

Altogether, these new data thus reveal mechanistic insights about where and at which step actin dynamics and clearance are important for ESCRT-III recruitment.

3. The correlative light-scanning electron microscopy in Fig 3b is very nice, but the conclusions would have been strengthened if quantifications could be provided.

Quantifications have now been added (p. 9). Numerous extensions and blebs in the region around the midbody were observed in 16/19 MICAL1-depleted bridges analyzed by SEM, but never observed in control cells (0/23 bridges).

4. The authors state that “a strong delay in cytokinetic abscission was observed after MICAL1 depletion”. This seems to be an overstatement given that more than 70% of the MICAL1 depleted cells show a very minor delay in completed abscission (Figs. 2d and 3d). Only a subpopulation of cells shows profound abscission delays upon MICAL1 knockdown. The authors should discuss what this might mean.

We agree with the Reviewer. “Strong delay” was referring to the fact that the two abscission time distributions were different with p values of $p = 0.000$ (KS test). We have removed “strong” in the text (p. 7) and now indicate in the discussion that additional pathways such as Rab35/OCRL and Rab11/p50RhoGAP (or other pathways) likely cooperate to regulate such an important step in cytokinesis (p. 20).

5. The Rab35-MICAL1 binding data in Fig 4d are not very convincing. There seems to be more of the Q67L than of the S22N mutant in the input, which could account for the (small) difference in detected binding between the two mutants and MICAL1. The ITC data showed as Supplementary information are more convincing, so perhaps these could be moved into the main figure.

We now provide other CoIP experiments (new Fig. 4d), which confirm ITC experiments and initial findings: Rab35 interacts with MICAL1 in a GTP-dependent manner, both in vitro and in cells.

Reviewer #3 (Remarks to the Author):

This is an interesting paper, describing the mechanism by which MICAL1 contributes to actin cytoskeleton disassembly at the abscission site during cytokinesis. The findings are important, and concern a fundamental biological process, which is of interest to the general readership of Nature Cell Biology. However, there are several issues that need to be addressed before this paper can be accepted. I would like to see a revised version that takes into account the criticism below.

We thank the Reviewer for this positive feedback about our work.

First, the major findings making this an important paper for NCB:

- 1) MICAL1 was shown to localize to the abscission site during cytokinesis, where it is necessary for successful abscission.
- 2) Abscission defects upon MICAL1 depletion were shown to result from F-actin accumulation at the intracellular bridge, consistent with MICAL1’s established role as an oxidation-dependent F-actin depolymerizing enzyme.
- 3) MICAL1 localization to abscission sites was shown to depend on an interaction with Rab35. A C-terminal helical domain in MICAL1 was shown to mediate this interaction.
- 4) A 3.3Å resolution crystal structure of the C-terminal helical domain of MICAL1 was determined, revealing a fold consisting of a 3-helix bundle. The structure was then used to design mutations that revealed a potential binding interface for Rab35.
- 5) Finally, TIRF microscopy in a microfluidics chamber was used to demonstrate that

MICAL1-mediated oxidation of F-actin leads to depolymerization from the barbed and pointed ends of the filament, and not through filament severing as previously proposed. While these are important results, several concerns need to be addressed before this paper can be accepted. My major concern is that, overall, the findings are meritorious enough to deserve publication in NCB, and yet the manuscript frequently overstates the novelty of the findings, either by not properly recognizing previous work or by over-interpreting the results.

1. In my opinion, the most important and novel contribution of this work, and the one that brings it up to the level of a high-profile journal such as NCB, is the finding that MICAL1 disassembles F-actin at the intracellular bridge during abscission, and under the control of Rab35. These points are convincingly demonstrated.

We agree with this point and thank the Reviewer for this comment.

However, the way the paper is written, the reader may get the impression that the authors came to this conclusion in the absence of previous knowledge. This is not the case. It was clearly established that MICAL1 and Rab35 interact (Fukuda et al. 2008 and Kobayashi et al., 2014). Moreover, the role of Rab35 in clearing actin filaments during abscission was also well established prior to this work (see Dambournet et al., Nat Cell Biol. 2011 and Prekeris, Cell Res. 2011), and latrunculin treatment (as done here) was already known to rescue cytokinesis (Dambournet et al., Nat Cell Biol. 2011). Finally, the mechanism described in this paper was explicitly proposed in a review paper that is not even cited (Chaineau et al. 2013). Specifically, the authors of this review wrote “Rab35 may additionally recruit MICAL1 to the intracellular bridge to further promote actin disassembly, allowing for cytokinesis”. The other papers are cited here, but within a context that does not allow the reader to recognize that the MICAL1-Rab35 interaction was already well established, as well as the role of Rab35 in actin disassembly during cytokinesis. All this prior knowledge should be properly acknowledged right in the Introduction, as the preamble to this work, and the authors should more humbly acknowledge that what is done here is to confirm/expand on these findings/proposals and establish a more detailed mechanism of the MICAL1-Rab35 interaction and its functional role.

It was not our intention to under-cite previous work. The role of Rab35 and Rab11 in regulating actin dynamics during cytokinesis was actually stated in the original manuscript (introduction, p.4). The 2-hybrid interaction between several Rabs and MICAL1 (Fukuda 2008) was also cited in the discussion. We agree that it is indeed well established that Rab35 directly interacts with MICAL-L1 (MICAL-Like 1, = Kobayashi 2014), a scaffolding protein in membrane traffic. However, to our knowledge, there has been no proof of a direct interaction between Rab35 and MICAL1. Finally, note that the rescue experiment with latrunculin has been described previously in OCRL-depleted cells (our paper Dambournet et al. NCB 2011), but latrunculin has not been investigated in MICAL1-depleted cells.

To address the Reviewer’s comments, we have reorganized the introduction and discussion. We now describe already in the introduction that Rab35 has been found to interact with MICAL1 by 2-hybrid experiments (Fukuda 2008) and by coIP (Deng 2016, published while the manuscript was being reviewed). We also cite in the introduction the Review from our

colleagues Chaineau et al. 2013 and their hypothesis that Rab35 together with MICAL1 might regulate cytokinesis.

2. One important experiment within this first part of the paper is listed as “data not shown”, and concerns the 2-hybrid assay revealing that the Rab35 binding site is contained within the MICAL1 fragment 918-1067. First of all, I am not sure whether NCB accepts “data not shown”. But more importantly, why not show this data, which is new and crucial to this paper? This fragment is subsequently used in biochemical experiments and in the crystal structure.

In the original version of the manuscript, we had shown the results of ITC experiments demonstrating that the 879-1067 and 918-1067 fragments have similar affinity for GppNHp-Rab35, which we believed was more informative (Table S1). As requested, in the revised manuscript, we provide in addition the 2-hybrid experiment with the 918-1067 fragment (new Fig. S1g).

3. However, the single most overstated portion of the paper is the structure of the C-terminal helical domain of MICAL1 and connected biochemical experiments. The structure, consisting of a 3-helix bundle, is described several times in the manuscript as a “novel” fold. First of all, I am rather sure that such claims of novelty are not endorsed by NCB. Moreover, the claim is unsupported. Strictly speaking, this is not a particularly reliable structure. The resolution is low (3.3 Å), the average B-factor is high (121 Å², and please give the units in Table S2), and many regions of the structure are disordered (the N- and C-terminal ends, as well as the linkers between helices). None of this is explained in the text, as it should, whereas sufficient space is found for several claims of novelty. The exact fragment crystallized and the regions visualized in the structure should be specified, along with other key parameters of the structure, such as the resolution.

Although of low resolution, the structure was carefully determined and validated by SAXS experiments. The structure based mutational analysis gave us confidence since the residues mutated on the surface based on our assignment of the residues did not change the fold (MALS and SAXS in Fig. S5) but changed Rab35 binding affinities for some mutants (ITC in Table S1) and allowed us to identify the surface involved in Rab binding. While our manuscript was being reviewed, a 2.8Å structure of MICAL1 RBD bound to two Rab10 was released (Rai and al., 2016). This structure (PDB ID 5LPN) confirms that our low resolution structure is correct (rmsd between our structure and 5LPN is 1.8 Å for 101 residues) and that regions connecting the helices are not well resolved in density since they are likely flexible (the H1-H2 loop connection is in a different orientation stabilized by Rab10 binding in 5LPN while the H2-H3 loop connection is also disordered in 5LPN). We now refer to this publication in the results section of the paper (p. 14).

In the revised manuscript, we have removed the claim of a novel fold and insisted however on the particularities of the RBD domain, which is not a helix bundle but instead forms a flat curved helical domain exposing mainly two extensive surfaces (*please, see detailed answer below*). This is an important point since this domain architecture is directly linked to the ability of the domain to bind Rab proteins.

We have now added the missing units in Table S2 and we included a sentence for the exact fragment crystallized and loops missing in the structure under the Table S2, as requested.

What is more, similar 3-helix bundles are extremely abundant, perhaps with a twist here and there or a shorter/longer alpha-helix, but definitely this is not a novel fold. In addition to the examples of 3-helix structures given in Figure S3, there are many other structures that contain related helical bundles as part of their core domains (vinculin, the spectrin repeat, the BAR domain, etc.). I actually found three helices in a BAR domain structure that look very similar to those shown in Figure 5. If anything, such a strong claim of folding novelty should be backed by a search using fold recognition software such as Dali or DEJAVU.

As stated in the previous point, our intention was to insist in the specific features of this domain. We have now removed from the text the claim of novel fold, re-wrote this section to clearly describe this structural domain to the reader (p. 13). The first and third helices of this domain do not interact but make specific interactions with the central anti-parallel second helix so that a flat domain with two opposite surfaces is formed. What is unusual is that we show that this three-helical domain is a monomer that exposes these two flat surfaces to partners such as Rab proteins. This differs from proteins that use such three-helix motifs to oligomerize such as BAR domains, or to interact with other structural elements of the protein. Since this C-ter domain was predicted as containing a long coiled-coil (potentially able to bind multiple Rabs), this fold was not expected.

In fact, we described in Figure S3 of the original manuscript the result of a search with the fold recognition software PDBeFold and indicated what differs in the C-ter structure compared to previous protein structures of highest similarity using this software. This information remains in the manuscript as Fig. S3.

In order to stay close to the central story, we mainly describe in the text the structural features that allow Rab binding, which was key for us to identify mutants unable to bind Rab35. We thus modified the Figure 5 of the revised manuscript by swapping the sequence alignment (previously as Fig. 5b and now in Fig. S4) for the results of the localization in cytokinetic bridges of point mutants (previously as Fig. S6d).

4. Describing the structure as a “three-helix sheet” can be confusing, since sheet is used for beta-strands.

It is the particularity of this domain that it doesn't really form a compact bundle but a sheet.

Moreover, the definition of ‘coiled-coil sheet’ has been previously introduced in the literature such as : “coiled-coil sheets” – see figure 8 in the paper: The structure of a-helical coiled coils by Andrei N. Lupas and Markus Gruber, *Adv Protein Chem.* 2005;70:37-78. This reference is now added to the discussion where we present this domain (p. 21).

5. Based on the structure, MALS and AUC experiments, the authors conclude that MICAL1 is monomeric. While the data for the particular fragment analyzed here is consistent with this

conclusion, the authors should be careful not to generalize this conclusion to the full-length protein that they did not analyze. As a matter of fact, 3-helix folds are often found in proteins that form dimers, and Rab effectors also tend to be helical dimers.

We agree that we don't have experimental data to confirm the oligomerization state of full length MICAL1, thus we cannot exclude that full length MICAL1 may form dimers upon activation. In the text, we now make clear that the C-terminal region involved in Rab binding is monomeric (p. 13).

6. The mutagenesis to define the Rab35-binding site based on the structure, while informative, should be presented as tentative. Let's recall that several parts of the structure are missing, including the loops that connect the three helices and several C-terminal amino acids (which other parts of the paper suggest is important for Rab35 binding). Specifically, the sentence "The surface responsible for Rab35 binding was identified since only one of the two mutants (mutant S1) was able to bind Rab35 as wild type" should be revised. Indeed, the results suggest that amino acids in mutant S2 may be involved in Rab35 binding (or that this multi-site mutation changes the structure of the binding site), but this does not strictly "identify the binding surface" as stated. Also, check the sentence "The surface [...] is in large part composed of a core of exposed hydrophobic residues surrounded by charged residues (Fig. 5d)". Fig. 5d does not really show much of an exposed hydrophobic core (and hydrophobic cores are not exposed by definition). The surface shown appears for the most part to be charged. Finally, the scale for the electrostatic surface representation must be given.

We do not agree that the mutant studies are tentative. We have chosen two sets of mutants targeting conserved exposed residues on either side of the flat domain. First, we show that six mutations of exposed residues on one of the surfaces do not change the affinity for Rab35, while SAXS and MALS studies indicate that the fold is not perturbed by these mutations (see Fig. S5 of the original manuscript). Second, we show that mutations on the other side destroy interaction with Rab35, while SAXS and MALS studies show that the fold is similar to that of the WT. These results are strengthened by single mutations on this surface that also impair binding to Rab35. This clearly indicates that only one surface must be involved in Rab35 binding and also it identifies where this binding site is located.

A recently published contribution described structures of MICAL RBD bound to different Rabs (Rai and al., 2016), although not Rab35. This study is in agreement with our structure and identifies two potential surfaces of interaction with Rab proteins, which correspond to the two surfaces we have mutated. Importantly, our study is of particular interest considering that no publication has yet described mutants that destroy the interaction of such a domain with Rab proteins. Another important contribution in our study is to characterize that for Rab35, only one of the two potential surfaces contributes to Rab35 binding. We also identified mutants to test in cells how MICAL1 gets recruited and activated.

The new Mical1:Rab10 complex structure (Rai and al., 2016) confirms the interaction site predicted by our mutational analysis. The structure shows that Rab10 specifically interacts with hydrophobic residues in the central part of the interaction site surrounded by charged

residues. We changed the Fig. 5c (previously Fig. 5d) as requested: it shows in a slightly different orientation the potential Rab35 binding site identified by the mutational analysis. The hydrophobic nature of this site is easier to see and we have now labeled hydrophobic residues. We also show the scale of the electrostatic potential.

7. The TIRF microscopy experiments were apparently performed in the absence of actin monomers. Since monomers add faster at the barbed end, it may be important to test how the presence of monomers (as it is the case in cells) affects barbed end depolymerization by MICAL1-mediated oxidation.

Previous work from independent laboratories using bulk assays (e.g. Hung et al. Nature 2010; Zucchini et al. Arch. Biochem. Biophys. 2011) already demonstrated that the presence of monomers is not enough to prevent depolymerization by MICAL.

It is also unclear why filaments on a cover slip appear to depolymerize only (or much faster) from the barbed end whereas in the microfluidics chamber they depolymerize from both ends. Since actin depolymerization in the absence of monomers is thought to occur primarily at the pointed end, it may be important to clarify if MICAL1 has a specific preference for the barbed end (in the presence/absence of monomers).

We apologize if this point was unclear. Filaments depolymerize faster from both ends, in both experimental configurations. We used the microfluidics setup to quantify the effect on the pointed end because it is more accurate than with filaments anchored to the coverslip (as mentioned in our manuscript, multiple anchoring points generate short pauses that add “noise” when measuring depolymerization rates. This problem is particularly acute for slower rates). We have modified the text to clarify this point.

Moreover, the conclusion that MICAL1 does not induce severing also appears excessive, since only the catalytic, monooxygenase/FAD domain was used in these experiments. For instance, Alqassim et al. 2016 reported that the adjacent CH domain is important for the enzymatic activity, and could also participate in F-actin binding, which could change the binding specificity of the enzyme along the filament.

In Fig. S7 of the original manuscript (now S7a-b), we used the FAD-CH-LIM and we did not observe severing either. Similarly, no severing was observed when full length MICAL1 was activated by Rab35-GTP (Fig. 6F).

Moreover, why would oxidation promote dissociation of terminal subunits but not breakage along the length of the filament? These experiments may also be lacking a key control – depolymerization induced by H₂O₂. Note that in addition to targeting substrates, MICAL1 is also known to produce H₂O₂ (Nadella et. al. 2005), which could non-specifically oxidize actin filaments.

We report here enhanced off-rates at the ends of the filaments as a consequence of their oxidation by MICAL1. This is consistent with the idea that MICAL-induced oxidation of filaments weakens the interaction between subunits within the filament. This is also the

interpretation proposed in Grintsevich et al. NCB 2016, and consistent with their observation: these filaments are easier to fragment, but thermal fluctuations are not enough (over our time scales, at least) and additional factors are required, such as mechanical stress (pipetting filament solutions) or severing proteins (cofilin). The strong illumination of fluorescently labelled filaments is also known to induce severing. We discuss these aspects in the discussion section of our manuscript (p. 21).

As reported by Hung et al. Science 2011 (Supp Mat and Supp Fig S1), MICAL does not alter actin dynamics through H₂O₂ production. To confirm this point, we monitored the barbed end depolymerization of ADP-F-actin in the presence of 40 mM H₂O₂, and found no significant difference with depolymerization in standard buffer. We now report these results in our new Fig. S7e.

We verified the potency of our H₂O₂ by monitoring the elongation of filaments from 1 μM G-actin incubated with 40 mM H₂O₂ for 20 minutes at 20°C and found that, in agreement with previous reports (DalleDonne et al. Biophys J 1995), H₂O₂ oxidizes G-actin and hinders polymerization: filaments elongated from H₂O₂-treated monomers at 2.4 ± 0.3 sub/s (Std Dev, N=12), compared to 7.8 ± 0.8 sub/s (Std Dev, N=12) in control experiments. This has been added in the Figure legend of new Fig. S7e.

Minor:

1. Labels too small in some of the figures.
2. Correct IR1055E in the legend to Figure S2.

These issues have been addressed. In particular, the size of the labels has been increased in Fig. 1 (a-c), Fig. 2 (a, c, d, e), Fig. 3 (b-d), Fig. 4 (a-i), Fig. 6e, Fig. S1 (d, g), Fig. S6 (a-b).

REVIEWERS' COMMENTS:

Reviewer #1 (Remarks to the Author):

Summary of revised manuscript:

This manuscript reports several major findings of interest to readers of the journal:

1. MICAL localizes to sites of cell abscission during cytokinesis where it is needed for successful abscission through removal of F-actin from the intracellular bridge.
2. MICAL localization depended on its interaction with Rab35 through a C-terminal helical domain.
3. MICAL can depolymerize filaments from either their pointed or barbed ends when these filaments are held in a flow chamber tethered at one end.

Comments:

The authors have attempted to address all of the major issues that were brought forward in the initial reviews. The more careful interpretations of some experiments offered by the authors along with the possibility of some alternative explanations satisfy most of the original criticisms.

The authors are commended for obtaining the *Drosophila* MICAL for comparative studies on its severing ability. Although both MICAL-1 and *Drosophila* MICAL did not sever by themselves in the assays performed herein, there remains a difference between the methods used in this paper and those used by Grintsevich et al, in which filament severing by MICAL alone also was reported. The Grintsevich et al. TIRF assay used F-actin that was trapped in a PEG matrix on the slide and was untethered. Thus, the conditions are not identical between the papers in which different results are shown.

The authors do believe that the end-only depolymerization by MICAL is the mechanism of subunit loss and that severing is an artifact of some other induced stress on filaments, such as pipetting.

Alternatively, it is also possible that MICAL-induced severing might only occur on filaments with a large degree of flexibility and that holding them linear in a flow cell through end tethering or using inactive myosin for side binding inhibits the filament flexibility required. An easy way to get around this problem is to just state that "under our conditions, MICAL has no severing activity." Given that MICAL-induced oxidation of actin makes it much more susceptible for cofilin-mediated severing (ref 26), direct severing versus depolymerization by MICAL alone is probably not terribly important physiologically.

I have no problems with the explanations offered in the discussion regarding why the differences were observed, but the authors should also acknowledge that preventing or decreasing filament bending in the assays used here might also impact MICAL's ability to sever.

It would be easy to accommodate the different results by the following changes:

Abstract, line 10: Moreover, we show that under our conditions, MICAL1 does not act as a severing protein but instead induces F-actin depolymerization from both ends.

Page 5, line 5: Surprisingly, single filament assays using end or side bound tethers, demonstrate that MICAL-induced weakening of F-actin primarily induces rapid depolymerization from both ends rather than filament severing previously reported for both tethered²³ and untethered²⁶ filaments.

Page 21, line 13: An unexpected result of the present study is the finding that MICAL1 is not an F-actin severing protein under our conditions. We confirmed our observations using *Drosophila* MICAL (supplementary Fig. 7c-d) and verified that MICAL is not altering actin dynamics through H₂O₂ production (Supplementary Fig. 7e), as previously reported²³. Enhanced depolymerization is consistent with the notion that MICAL1 oxidation weakens longitudinal interactions between F-actin subunits. We expect this weakening to make filaments more sensitive to fragmentation-inducing conditions, and thus believe that the initially observed severing²³ was provoked by filament-surface interactions, high fluorescent labeling fractions and/or illumination conditions. However, the possibility remains that end-tethering of filaments in a flow cell, or the degree of side-tethering of filaments, constrains their bending motions needed to observe MICAL-induced severing on its own. In fact...

Reviewer #2 (Remarks to the Author):

The authors have successfully addressed the concerns I had, and I am happy to recommend publication of this revised manuscript.

Reviewer #3 (Remarks to the Author):

This review is an update on my previous review for NCB. The authors have addressed most of my concerns. The paper appears appropriate for Nature Communications. It must be pointed out, however, that some of the work presented here is not entirely novel. This includes the structural work, and well as the involvement of MICAL isoforms in cytokinesis. Thus, Liu et al., JBC 2016 already found that MICAL3 is involved in cytokinesis, and Rai et al., eLIFE 2016 determined a series of structures, including a complex of the C-terminal region of MICAL1 with Rab10. Nevertheless, the authors acknowledge these studies, and the work presented here implicates a different MICAL isoform in cytokinesis (MICAL1), and is more complete in that it addresses this topic using a comprehensive, cellular, biochemical and structural approach. The findings are important, and concern a fundamental biological process, which is of interest to the general readership of Nature Communications. A minor criticism - the paper would benefit from a more detailed discussion of MICAL function, including its N-terminal domain for which several structures have been determined (not referenced). In other words, how do the non-catalytic domains influence/regulate MICAL's activities?

Response to Reviewers

“Oxidation of F-actin controls the terminal steps of cell division”

We thank the three Reviewers for their very positive comments and for suggested experiments. We have provided additional experiments and Figures as well as explanations in the revised manuscript to address all their comments. In the revised manuscript, we have reorganized and shortened the introduction and the discussion in order to comply with *Nature Communications* guidelines.

New experimental data have been added as. 3e, 3f, 4d, Supplementary Fig. S1f, S1g, S2j, S6d, S7c, S7d, S7e, S8a and S8b.

We provide below a full response to each comment raised by the Reviewers. We believe that the added experimental data and Reviewer's suggestions helped us to reinforce the conclusions of the manuscript.

Reviewer #1 (Remarks to the Author):

Summary: This manuscript presents data that explores the role of MICAL1 in cell abscission, the final step of cell division. MICAL1 is an NADPH oxidase that targets actin oxidation and one hypothesis being tested in this manuscript is that MICAL1 is required to remove actin in the final stages of abscission. The mechanism by which MICAL1 gets localized to the bridge region of the abscission zone and the mechanism by which MICAL1 depolymerizes the F-actin in this region are also addressed, as is the relationship between the localization of MICAL1 and CHMP4B, a component of the ESCRT complex previously shown to co-localize with spastin, a microtubule severing protein needed to clear microtubules from the bridge region before abscission. It was previously known that MICAL1 interacted with Rab35 and herein the authors show this interaction is essential for MICAL1 localization to the cytokinetic bridge. Furthermore they identify the specific region of MICAL1 involved in Rab35 binding, discovering a novel three helix interaction in the process. They show that silencing MICAL1 delays the average time to abscission by quite a bit, although about 40-50% of cells do seem to undergo abscission in the normal time frame, suggesting that there are probably some redundant mechanisms at play in this process, a point not discussed in the manuscript even though additional evidence suggesting this is presented. The most surprising result of the manuscript is the finding that MICAL1 has no F-actin severing activity, as has been reported for *Drosophila* MICAL, but rather that its activity increases the rate of filament depolymerization from both ends through an NADPH dependent step, presumably the oxidation of the met44 of actin. This was shown most dramatically by an excellent experiment in which fluorescently tagged oxidized actin was elongated with fluorescently tagged normal actin and its rate of depolymerization followed in the absence of MICAL1. The filaments depolymerized slowly until the zone of oxidized subunits was reached and then rapid depolymerization ensued. The authors go to great lengths to show that their results hold up under their favored depolymerization conditions (F-actin that is end anchored) as well as the identical conditions used by others to show severing (linked by a dead myosin II to the surface).

Critique: This is an excellent paper that offers insight into one mechanism of actin removal from the cytokinetic bridge preceding abscission. The manuscript is well written, the science is sound, the methods are appropriate and the results are convincing.

We thank the Reviewer for this very positive evaluation of our work.

However, there are multiple ways for abscission to occur and the authors own evidence clearly supports this assumption:

1. Only 62% of cytokinetic bridges containing the ESCRT complex component CHMP4B have detectable pools of MICAL
2. Depletion of MICAL1 only entirely blocked abscission in 4% of cells (3 fold more than the control cell population, but not terribly impressive if MICAL1-dependent actin disassembly was the only mechanism for the actin removal to occur).
3. Although the average abscission time increased in MICAL1 silenced cells, about 40-50% of the cells completed abscission at nearly identical times as the wild type (control) population. Thus a bit more discussion on cooperative or redundant mechanisms would be appropriate.

We now provide a revised discussion specifically addressing this point (p. 20). In particular, we mention that additional pathways such as Rab35/OCRL and Rab11/p50RhoGAP (or other pathways) likely cooperate to regulate such an important step in cytokinesis.

In dividing cells, cofilin is also enriched within the cytokinetic bridges and may provide an alternative (or indeed even a cooperative) mechanism for actin removal. Work recently published by Grintsevich et al (NCB July 2016) demonstrates MICAL oxidized actin actually recruits cofilin to enhance the rapid depolymerization of the actin. Although MICAL-induced severing is also reported in the Gritsevich et al paper, the rapid loss of actin following cofilin addition to MICAL oxidized actin occurred in the absence of MICAL, showing that the oxidation lead to cofilin-enhanced depolymerization. While it is not incumbent upon these authors to provide a complete analysis of cofilin in their model, acknowledgement of other mechanisms, either redundant or cooperative, should be mentioned in the discussion. This is especially pertinent because the LatA rescue of abscission in MICAL depleted cells shown in this paper should only occur if the F-actin population is dynamic and this dynamics is usually dependent upon the presence of cofilin along with its entourage of proteins (coronin, Aip1, etc) that help in its recruitment and severing/depolymerizing functions.

Cofilin has a crucial role in furrow ingression by regulating actin ring dynamics. To our knowledge, the localization of cofilin later in cytokinetic bridges has not been reported. Our unpublished data indicate that a pool of cofilin is indeed detected in bridges, suggesting that MICAL1 and cofilin might cooperate during abscission. Since cofilin has a crucial role during furrow ingression, it is however technically challenging to test this hypothesis experimentally. We nevertheless discuss this interesting possibility in the light of the Grintsevich et al. paper on p. 21, as suggested.

With regard to newly proposed mechanism of F-actin depolymerization from either end by MICAL without severing (which I think is very clear and nicely demonstrated here) it should be noted that the previously published papers showing the severing mechanism utilized *Drosophila* MICAL and not mammalian MICAL1 used here. The authors' suggest that severing observed in these other assays may be artefactual due to photocleavage of oxidized fluorescent actin. While this may be the explanation, it is also possible that *Drosophila* MICAL has severing activity not found in mammalian MICALs. Thus the authors should obtain an aliquot of the *Drosophila* MICAL for a direct comparison. Otherwise, readers will be left to wonder if the other published studies are incorrect or if there are real differences between MICAL activities from different sources. This study is critical for publication of the paper in NCB because it directly contradicts some aspects of other work recently published. Indeed if the *Drosophila* MICAL does not sever at all, some results of the previous papers would need to be reinterpreted.

To extend our conclusions, we have confirmed our data using *Drosophila* MICAL in our own microfluidic set up and time scale, as requested. We used the same construct as in Hung et al. Science 2011 (dMICAL aa 1-669) and did not observe severing activity (new Fig. S7c-d). However, depolymerization from both ends was observed, as seen in our previous experiment using human MICAL FAD. We note that the latest report from the same group (Grintsevich et al. NCB 2016, which was published while our manuscript was being reviewed) now shows that severing of MICAL-oxidized filaments only occurs when filaments are stressed (pipetting) or exposed to cofilin. Altogether, we conclude that MICAL-dependent oxidation of F-actin weakens the filaments and that the primary consequence of this weakening is an enhanced depolymerization, not severing (discussion p. 21).

Minor corrections:

Page 4, line 2 and 3: end the sentence after microtubules- the first word on line 2- the remainder is redundant.

Page 4, line 7: The recruitment of the phosphatase to deplete PIP2 is clear, but the cause and effect conclusion of "therefore prevents F-actin accumulation in late..." is not so clear. It sounds like the authors are saying that F-actin cannot assemble in the absence of PIP2 which I am sure is not what the authors intended to imply. The wording here should be revised.

Page 8, last line and top of page 9: phalloidin is certainly used as a marker for F-actin but phalloidin does not bind F-actin saturated with cofilin- and if cofilin is preferentially recruited to MICAL oxidized actin, as work from Grintsevich et al implies, the loss of MICAL might reduce the cofilin recruitment which might make the filaments easier to label with phalloidin, accounting for some of the increase observed. Do the authors know if F-actin oxidation by MICAL affects phalloidin binding? Are the increased numbers of membrane bulges and extensions perhaps due to recruitment of Arp2/3 or formins to the midbody in the absence of MICAL oxidation of F-actin? Certainly the LatA effects described on page 9 indicate a dynamic actin pool in this region.

Page 10, last line: suggested revision ... at the abscission site was partially, but significantly, rescued by

Page 12, last paragraph, second sentence: suggested change: Whereas the C-terminal... with Rab35Q67L, removing the last...

Page 17, line 12: I believe 6e should be the referenced figure, not 6a as listed.

We modify the text, as requested.

In addition, we now provide evidence that indeed the p34^{Arc} subunit of Arp2/3 is clearly detected in MICAL1-depleted bridges (new Fig. S8b and discussion p. 20), confirming the presence of a dynamic pool of actin in bridges in this condition.

In addition, to rule out that potential interactions with cofilin would alter the original quantifications, we independently measured an increase of GFP-actin in bridges (using a genome-edited cell line expressing endogenous levels) in cytokinetic bridges after MICAL1 depletion (new Fig. S1f). Quantifications using labelled phalloidin and GFP-actin gave very similar results.

Reviewer #1 (Remarks to the Author, final requests):

Summary of revised manuscript:

This manuscript reports several major findings of interest to readers of the journal:

1. MICAL localizes to sites of cell abscission during cytokinesis where it is needed for successful abscission through removal of F-actin from the intracellular bridge.
2. MICAL localization depended on its interaction with Rab35 through a C-terminal helical domain.
3. MICAL can depolymerize filaments from either their pointed or barbed ends when these filaments are held in a flow chamber tethered at one end.

Comments:

The authors have attempted to address all of the major issues that were brought forward in the initial reviews. The more careful interpretations of some experiments offered by the authors along with the possibility of some alternative explanations satisfy most of the original criticisms.

The authors are commended for obtaining the *Drosophila* MICAL for comparative studies on its severing ability. Although both MICAL-1 and *Drosophila* MICAL did not sever by themselves in the assays performed herein, there remains a difference between the methods used in this paper and those used by Grintsevich et al, in which filament severing by MICAL alone also was reported. The Grintsevich et al. TIRF assay used F-actin that was trapped in a PEG matrix on the slide and was untethered. Thus, the conditions are not identical between the papers in which different results are shown.

The authors do believe that the end-only depolymerization by MICAL is the mechanism of subunit loss and that severing is an artifact of some other induced stress on filaments, such as pipetting. Alternatively, it is also possible that MICAL-induced severing might only occur on filaments with a large degree of flexibility and that holding them linear in a flow cell through end tethering or using inactive myosin for side binding inhibits the filament flexibility required. An easy way to get around this problem is to just state that "under our conditions, MICAL has no severing activity." Given that MICAL-induced oxidation of actin makes it much more susceptible for cofilin-mediated severing (ref 26), direct severing versus depolymerization by MICAL alone is probably not terribly important physiologically.

I have no problems with the explanations offered in the discussion regarding why the differences were observed, but the authors should also acknowledge that preventing or decreasing filament bending in the assays used here might also impact MICAL's ability to sever.

It would be easy to accommodate the different results by the following changes:

Abstract, line 10: Moreover, we show that under our conditions, MICAL1 does not act as a severing protein but instead induces F-actin depolymerization from both ends.

Page 5, line 5: Surprisingly, single filament assays using end or side bound tethers, demonstrate that MICAL-induced weakening of F-actin primarily induces rapid depolymerization from both ends rather than filament severing previously reported for both tethered²³ and untethered²⁶ filaments.

Page 21, line 13: An unexpected result of the present study is the finding that MICAL1 is not an F-actin severing protein under our conditions. We confirmed our observations using *Drosophila* MICAL (supplementary Fig. 7c-d) and verified that MICAL is not altering actin dynamics through H₂O₂ production (Supplementary Fig. 7e), as previously reported²³. Enhanced depolymerization is consistent with the notion that MICAL1 oxidation weakens longitudinal interactions between F-actin subunits. We expect this weakening to make filaments more sensitive to fragmentation-inducing conditions, and thus believe that the initially observed severing²³ was provoked by filament-surface interactions, high fluorescent labeling fractions and/or illumination conditions. However, the possibility remains that end-tethering of filaments in a flow cell, or the degree of side-tethering of filaments, constrains their bending motions needed to observe MICAL-induced severing on its own. In fact...

We are pleased that the Reviewer is satisfied by the new data added in the revised manuscript.

For the final version of the manuscript, we have taken into account all editorial changes requested by the Reviewer:

Abstract: “Moreover, in our experimental conditions, MICAL1 does not act as a severing enzyme, as initially thought, but instead induces F-actin depolymerization from both ends.”

Introduction: “Surprisingly, single filament assays using end or side-bound tethers demonstrate that the MICAL1-induced weakening of F-actin primarily induces rapid depolymerization from both ends rather than filament severing previously reported for both tethered²³ and untethered²⁸ filaments. »

Discussion: “An unexpected result is the finding that, in our experiments under different conditions, MICAL1 is not an F-actin severing enzyme, as initially thought²³, and that its primary effect is to enhance filament depolymerization from both ends.”

Discussion: “We expect this weakening to make filaments more sensitive to fragmentation-inducing conditions, and thus believe that the initially observed severing²³ was provoked by filament-surface interactions, different degree of bending, high fluorescent labeling fractions and/or illumination conditions.”

Reviewer #2 (Remarks to the Author):

In this manuscript Fremont and colleagues report that oxidation of F-actin controls cytokinetic abscission. The authors find that MICAL1, a flavoprotein monooxygenase known to oxidize F-actin on Met-44, is recruited to the midbody and abscission zone by the small GTPase Rab35. Interference with MICAL1 or its catalytic activity causes cytokinetic abscission delay, accompanied by increased actin levels in the cytokinetic bridge. Recruitment of ESCRT-III, thought to mediate membrane scission, is impaired under these conditions, and recruitment can be restored with the actin poison Latrunculin A. Using a microfluidics assay the authors find that MICAL1 induces actin depolymerization from both ends and does not function as an actin-severing enzyme as previously proposed. They solve the crystal structure of the Rab35-binding region of MICAL1 at 3.3 Å and show that it consists of a curved sheet of three helices. Evidence is presented that MICAL1 exists in an autoinhibited conformation which is relieved upon Rab35 binding.

The technical standard of this manuscript is excellent, and most of the conclusions are well supported by the data. Collectively these findings provide substantial new insight into the regulation of actin dynamics during cytokinetic abscission and should be of wide interest.

We thank the Reviewer for this very positive evaluation of our work.

Specific comments:

1. The structure of the manuscript is a bit strange. The authors start with localization and functional studies of MICAL1, and later they identify – seemingly out of nowhere - MICAL1 as an interactor of Rab35 in a yeast two-hybrid screen for Rab35 effectors. Surely this study must have started with the two-hybrid screen?

The Reviewer is right, but we wanted to centre the paper on the main finding: the role of MICAL1 in F-actin oxidation during cell division. Then, to unravel how Rab proteins recruit MICAL1 and activate the redox activity. To address the Reviewer's comment, we have now cited what was known about MICAL1/Rabs interaction in the literature in the introduction (p. 4). We then characterize in details in the second part of the manuscript the relevance of the Rab35/MICAL1 interaction that we now demonstrate to be direct.

2. The weakest conclusion of the manuscript is the notion that MICAL1-mediated actin depolymerization is required for ESCRT-III recruitment. It is interesting that CHMP4B was correctly recruited in only 25% of the cytokinetic bridges in MICAL1 depleted cells (Fig 3e), but even in control cells only a minority of the cytokinetic bridges showed recruitment of CHMP4B. How can this be explained? And how would F-actin prevent ESCRT-III recruitment? Does stabilization of F-actin with Jasplakinolide prevent ESCRT-III recruitment?

We have rewritten the corresponding text to make it clearer, as these figures actually referred to CHMP4B at the abscission (secondary) site specifically and not at the whole bridge, which should reassure the Reviewer. Actually, 85% of the control bridges contain

CHMP4B (either at the midbody or at the abscission site), i.e. in the great majority of the bridges, as expected (p. 10 and Fig. 3e-f).

In order to gain insights into the role of MICAL1-mediated actin depolymerization in ESCRT-III recruitment, we now provide additional quantifications of ESCRT-III recruitment at the midbody only vs. midbody + abscission site, both in control- and in MICAL1-depleted cells (new Fig. 3e and 3f). CHMP4B was correctly recruited at the abscission site in only 23% of the bridges in MICAL1-depleted cells, in contrast to 39% in control cells (Figure 3e). Concomitantly, an increase of bridges with CHMP4B localized only at the midbody was observed after MICAL1 depletion (Figure 3e). Altogether, these results suggest that clearing F-actin from the bridge does not perturb initial ESCRT-III recruitment at the midbody but its subsequent recruitment at the abscission site (p. 10).

Finally, we experimentally tested the effect of Jasplakinolide on ESCRT-III recruitment, as requested. Interestingly, Jasplakinolide treatment reduced ESCRT-III recruitment to the abscission site (new Fig. S8a and discussion p.20), suggesting that chemical stabilization of actin also perturbs the recruitment of the abscission machinery.

Altogether, these new data thus reveal mechanistic insights about where and at which step actin dynamics and clearance are important for ESCRT-III recruitment.

3. The correlative light-scanning electron microscopy in Fig 3b is very nice, but the conclusions would have been strengthened if quantifications could be provided.

Quantifications have now been added (p. 9). Numerous extensions and blebs in the region around the midbody were observed in 16/19 MICAL1-depleted bridges analyzed by SEM, but never observed in control cells (0/23 bridges).

4. The authors state that “a strong delay in cytokinetic abscission was observed after MICAL1 depletion”. This seems to be an overstatement given that more than 70% of the MICAL1 depleted cells show a very minor delay in completed abscission (Figs. 2d and 3d). Only a subpopulation of cells shows profound abscission delays upon MICAL1 knockdown. The authors should discuss what this might mean.

We agree with the Reviewer. “Strong delay” was referring to the fact that the two abscission time distributions were different with p values of $p = 0.000$ (KS test). We have removed “strong” in the text (p. 7) and now indicate in the discussion that additional pathways such as Rab35/OCRL and Rab11/p50RhoGAP (or other pathways) likely cooperate to regulate such an important step in cytokinesis (p. 20).

5. The Rab35-MICAL1 binding data in Fig 4d are not very convincing. There seems to be more of the Q67L than of the S22N mutant in the input, which could account for the (small) difference in detected binding between the two mutants and MICAL1. The ITC data showed as Supplementary information are more convincing, so perhaps these could be moved into the main figure.

We now provide other CoIP experiments (new Fig. 4d), which confirm ITC experiments and initial findings: Rab35 interacts with MICAL1 in a GTP-dependent manner, both in vitro and in cells.

Reviewer #2 (Remarks to the Author):

The authors have successfully addressed the concerns I had, and I am happy to recommend publication of this revised manuscript.

Reviewer #3 (Remarks to the Author):

This is an interesting paper, describing the mechanism by which MICAL1 contributes to actin cytoskeleton disassembly at the abscission site during cytokinesis. The findings are important, and concern a fundamental biological process, which is of interest to the general readership of Nature Cell Biology. However, there are several issues that need to be addressed before this paper can be accepted. I would like to see a revised version that takes into account the criticism below.

We thank the Reviewer for this positive feedback about our work.

First, the major findings making this an important paper for NCB:

- 1) MICAL1 was shown to localize to the abscission site during cytokinesis, where it is necessary for successful abscission.
- 2) Abscission defects upon MICAL1 depletion were shown to result from F-actin accumulation at the intracellular bridge, consistent with MICAL1's established role as an oxidation-dependent F-actin depolymerizing enzyme.
- 3) MICAL1 localization to abscission sites was shown to depend on an interaction with Rab35. A C-terminal helical domain in MICAL1 was shown to mediate this interaction.
- 4) A 3.3Å resolution crystal structure of the C-terminal helical domain of MICAL1 was determined, revealing a fold consisting of a 3-helix bundle. The structure was then used to design mutations that revealed a potential binding interface for Rab35.
- 5) Finally, TIRF microscopy in a microfluidics chamber was used to demonstrate that MICAL1-mediated oxidation of F-actin leads to depolymerization from the barbed and pointed ends of the filament, and not through filament severing as previously proposed. While these are important results, several concerns need to be addressed before this paper can be accepted. My major concern is that, overall, the findings are meritorious enough to deserve publication in NCB, and yet the manuscript frequently overstates the novelty of the findings, either by not properly recognizing previous work or by over-interpreting the results.

1. In my opinion, the most important and novel contribution of this work, and the one that brings it up to the level of a high-profile journal such as NCB, is the finding that MICAL1 disassembles F-actin at the intracellular bridge during abscission, and under the control of Rab35. These points are convincingly demonstrated.

We agree with this point and thank the Reviewer for this comment.

However, the way the paper is written, the reader may get the impression that the authors came to this conclusion in the absence of previous knowledge. This is not the case. It was clearly established that MICAL1 and Rab35 interact (Fukuda et al. 2008 and Kobayashi et al., 2014). Moreover, the role of Rab35 in clearing actin filaments during abscission was also well established prior to this work (see Dambournet et al., Nat Cell Biol. 2011 and Prekeris, Cell Res. 2011), and latrunculin treatment (as done here) was already known to rescue cytokinesis (Dambournet et al., Nat Cell Biol. 2011). Finally, the mechanism described in this paper was explicitly proposed in a review paper that is not even cited (Chaineau et al. 2013). Specifically, the authors of this review wrote “Rab35 may additionally recruit MICAL1 to the intracellular bridge to further promote actin disassembly, allowing for cytokinesis”. The other papers are cited here, but within a context that does not allow the reader to recognize that the MICAL1-Rab35 interaction was already well established, as well as the role of Rab35 in actin disassembly during cytokinesis. All this prior knowledge should be properly acknowledged right in the Introduction, as the preamble to this work, and the authors should more humbly acknowledge that what is done here is to confirm/expand on these findings/proposals and establish a more detailed mechanism of the MICAL1-Rab35 interaction and its functional role.

It was not our intention to under-cite previous work. The role of Rab35 and Rab11 in regulating actin dynamics during cytokinesis was actually stated in the original manuscript (introduction, p.4). The 2-hybrid interaction between several Rabs and MICAL1 (Fukuda 2008) was also cited in the discussion. We agree that it is indeed well established that Rab35 directly interacts with MICAL-L1 (MICAL-Like 1, = Kobayashi 2014), a scaffolding protein in membrane traffic. However, to our knowledge, there has been no proof of a direct interaction between Rab35 and MICAL1. Finally, note that the rescue experiment with latrunculin has been described previously in OCRL-depleted cells (our paper Dambournet et al. NCB 2011), but latrunculin has not been investigated in MICAL1-depleted cells.

To address the Reviewer’s comments, we have reorganized the introduction and discussion. We now describe already in the introduction that Rab35 has been found to interact with MICAL1 by 2-hybrid experiments (Fukuda 2008) and by coIP (Deng 2016, published while the manuscript was being reviewed). We also cite in the introduction the Review from our colleagues Chaineau et al. 2013 and their hypothesis that Rab35 together with MICAL1 might regulate cytokinesis.

2. One important experiment within this first part of the paper is listed as “data not shown”, and concerns the 2-hybrid assay revealing that the Rab35 binding site is contained within the MICAL1 fragment 918-1067. First of all, I am not sure whether NCB accepts “data not shown”. But more importantly, why not show this data, which is new and crucial to this paper? This fragment is subsequently used in biochemical experiments and in the crystal

structure.

In the original version of the manuscript, we had shown the results of ITC experiments demonstrating that the 879-1067 and 918-1067 fragments have similar affinity for GppNHp-Rab35, which we believed was more informative (Table S1). As requested, in the revised manuscript, we provide in addition the 2-hybrid experiment with the 918-1067 fragment (new Fig. S1g).

3. However, the single most overstated portion of the paper is the structure of the C-terminal helical domain of MICAL1 and connected biochemical experiments. The structure, consisting of a 3-helix bundle, is described several times in the manuscript as a “novel” fold. First of all, I am rather sure that such claims of novelty are not endorsed by NCB. Moreover, the claim is unsupported. Strictly speaking, this is not a particularly reliable structure. The resolution is low (3.3 Å), the average B-factor is high (121 Å², and please give the units in Table S2), and many regions of the structure are disordered (the N- and C-terminal ends, as well as the linkers between helices). None of this is explained in the text, as it should, whereas sufficient space is found for several claims of novelty. The exact fragment crystallized and the regions visualized in the structure should be specified, along with other key parameters of the structure, such as the resolution.

Although of low resolution, the structure was carefully determined and validated by SAXS experiments. The structure based mutational analysis gave us confidence since the residues mutated on the surface based on our assignment of the residues did not change the fold (MALS and SAXS in Fig. S5) but changed Rab35 binding affinities for some mutants (ITC in Table S1) and allowed us to identify the surface involved in Rab binding. While our manuscript was being reviewed, a 2.8Å structure of MICAL1 RBD bound to two Rab10 was released (Rai and al., 2016). This structure (PDB ID 5LPN) confirms that our low resolution structure is correct (rmsd between our structure and 5LPN is 1.8 Å for 101 residues) and that regions connecting the helices are not well resolved in density since they are likely flexible (the H1-H2 loop connection is in a different orientation stabilized by Rab10 binding in 5LPN while the H2-H3 loop connection is also disordered in 5LPN). We now refer to this publication in the results section of the paper (p. 14).

In the revised manuscript, we have removed the claim of a novel fold and insisted however on the particularities of the RBD domain, which is not a helix bundle but instead forms a flat curved helical domain exposing mainly two extensive surfaces (*please, see detailed answer below*). This is an important point since this domain architecture is directly linked to the ability of the domain to bind Rab proteins.

We have now added the missing units in Table S2 and we included a sentence for the exact fragment crystallized and loops missing in the structure under the Table S2, as requested.

What is more, similar 3-helix bundles are extremely abundant, perhaps with a twist here and there or a shorter/longer alpha-helix, but definitely this is not a novel fold. In addition to the examples of 3-helix structures given in Figure S3, there are many other structures that contain related helical bundles as part of their core domains (vinculin, the spectrin repeat,

the BAR domain, etc.). I actually found three helices in a BAR domain structure that look very similar to those shown in Figure 5. If anything, such a strong claim of folding novelty should be backed by a search using fold recognition software such as Dali or DEJAVU.

As stated in the previous point, our intention was to insist in the specific features of this domain. We have now removed from the text the claim of novel fold, re-wrote this section to clearly describe this structural domain to the reader (p. 13). The first and third helices of this domain do not interact but make specific interactions with the central anti-parallel second helix so that a flat domain with two opposite surfaces is formed. What is unusual is that we show that this three-helical domain is a monomer that exposes these two flat surfaces to partners such as Rab proteins. This differs from proteins that use such three-helix motifs to oligomerize such as BAR domains, or to interact with other structural elements of the protein. Since this C-ter domain was predicted as containing a long coiled-coil (potentially able to bind multiple Rabs), this fold was not expected.

In fact, we described in Figure S3 of the original manuscript the result of a search with the fold recognition software PDBeFold and indicated what differs in the C-ter structure compared to previous protein structures of highest similarity using this software. This information remains in the manuscript as Fig. S3.

In order to stay close to the central story, we mainly describe in the text the structural features that allow Rab binding, which was key for us to identify mutants unable to bind Rab35. We thus modified the Figure 5 of the revised manuscript by swapping the sequence alignment (previously as Fig. 5b and now in Fig. S4) for the results of the localization in cytokinetic bridges of point mutants (previously as Fig. S6d).

4. Describing the structure as a “three-helix sheet” can be confusing, since sheet is used for beta-strands.

It is the particularity of this domain that it doesn't really form a compact bundle but a sheet.

Moreover, the definition of ‘coiled-coil sheet’ has been previously introduced in the literature such as : “coiled-coil sheets” – see figure 8 in the paper: The structure of a-helical coiled coils by Andrei N. Lupas and Markus Gruber, *Adv Protein Chem.* 2005;70:37-78. This reference is now added to the discussion where we present this domain (p. 21).

5. Based on the structure, MALS and AUC experiments, the authors conclude that MICAL1 is monomeric. While the data for the particular fragment analyzed here is consistent with this conclusion, the authors should be careful not to generalize this conclusion to the full-length protein that they did not analyze. As a matter of fact, 3-helix folds are often found in proteins that form dimers, and Rab effectors also tend to be helical dimers.

We agree that we don't have experimental data to confirm the oligomerization state of full length MICAL1, thus we cannot exclude that full length MICAL1 may form dimers upon activation. In the text, we now make clear that the C-terminal region involved in Rab binding is monomeric (p. 13).

6. The mutagenesis to define the Rab35-binding site based on the structure, while informative, should be presented as tentative. Let's recall that several parts of the structure are missing, including the loops that connect the three helices and several C-terminal amino acids (which other parts of the paper suggest is important for Rab35 binding). Specifically, the sentence "The surface responsible for Rab35 binding was identified since only one of the two mutants (mutant S1) was able to bind Rab35 as wild type" should be revised. Indeed, the results suggest that amino acids in mutant S2 may be involved in Rab35 binding (or that this multi-site mutation changes the structure of the binding site), but this does not strictly "identify the binding surface" as stated. Also, check the sentence "The surface [...] is in large part composed of a core of exposed hydrophobic residues surrounded by charged residues (Fig. 5d)". Fig. 5d does not really show much of an exposed hydrophobic core (and hydrophobic cores are not exposed by definition). The surface shown appears for the most part to be charged. Finally, the scale for the electrostatic surface representation must be given.

We do not agree that the mutant studies are tentative. We have chosen two sets of mutants targeting conserved exposed residues on either side of the flat domain. First, we show that six mutations of exposed residues on one of the surfaces do not change the affinity for Rab35, while SAXS and MALS studies indicate that the fold is not perturbed by these mutations (see Fig. S5 of the original manuscript). Second, we show that mutations on the other side destroy interaction with Rab35, while SAXS and MALS studies show that the fold is similar to that of the WT. These results are strengthened by single mutations on this surface that also impair binding to Rab35. This clearly indicates that only one surface must be involved in Rab35 binding and also it identifies where this binding site is located.

A recently published contribution described structures of MICAL RBD bound to different Rabs (Rai and al., 2016), although not Rab35. This study is in agreement with our structure and identifies two potential surfaces of interaction with Rab proteins, which correspond to the two surfaces we have mutated. Importantly, our study is of particular interest considering that no publication has yet described mutants that destroy the interaction of such a domain with Rab proteins. Another important contribution in our study is to characterize that for Rab35, only one of the two potential surfaces contributes to Rab35 binding. We also identified mutants to test in cells how MICAL1 gets recruited and activated.

The new Mical1:Rab10 complex structure (Rai and al., 2016) confirms the interaction site predicted by our mutational analysis. The structure shows that Rab10 specifically interacts with hydrophobic residues in the central part of the interaction site surrounded by charged residues. We changed the Fig. 5c (previously Fig. 5d) as requested: it shows in a slightly different orientation the potential Rab35 binding site identified by the mutational analysis. The hydrophobic nature of this site is easier to see and we have now labeled hydrophobic residues. We also show the scale of the electrostatic potential.

7. The TIRF microscopy experiments were apparently performed in the absence of actin monomers. Since monomers add faster at the barbed end, it may be important to test how

the presence of monomers (as it is the case in cells) affects barbed end depolymerization by MICAL1-mediated oxidation.

Previous work from independent laboratories using bulk assays (e.g. Hung et al. Nature 2010; Zucchini et al. Arch. Biochem. Biophys. 2011) already demonstrated that the presence of monomers is not enough to prevent depolymerization by MICAL.

It is also unclear why filaments on a cover slip appear to depolymerize only (or much faster) from the barbed end whereas in the microfluidics chamber they depolymerize from both ends. Since actin depolymerization in the absence of monomers is thought to occur primarily at the pointed end, it may be important to clarify if MICAL1 has a specific preference for the barbed end (in the presence/absence of monomers).

We apologize if this point was unclear. Filaments depolymerize faster from both ends, in both experimental configurations. We used the microfluidics setup to quantify the effect on the pointed end because it is more accurate than with filaments anchored to the coverslip (as mentioned in our manuscript, multiple anchoring points generate short pauses that add “noise” when measuring depolymerization rates. This problem is particularly acute for slower rates). We have modified the text to clarify this point.

Moreover, the conclusion that MICAL1 does not induce severing also appears excessive, since only the catalytic, monooxygenase/FAD domain was used in these experiments. For instance, Alqassim et al. 2016 reported that the adjacent CH domain is important for the enzymatic activity, and could also participate in F-actin binding, which could change the binding specificity of the enzyme along the filament.

In Fig. S7 of the original manuscript (now S7a-b), we used the FAD-CH-LIM and we did not observe severing either. Similarly, no severing was observed when full length MICAL1 was activated by Rab35-GTP (Fig. 6F).

Moreover, why would oxidation promote dissociation of terminal subunits but not breakage along the length of the filament? These experiments may also be lacking a key control – depolymerization induced by H₂O₂. Note that in addition to targeting substrates, MICAL1 is also known to produce H₂O₂ (Nadella et. al. 2005), which could non-specifically oxidize actin filaments.

We report here enhanced off-rates at the ends of the filaments as a consequence of their oxidation by MICAL1. This is consistent with the idea that MICAL-induced oxidation of filaments weakens the interaction between subunits within the filament. This is also the interpretation proposed in Grintsevich et al. NCB 2016, and consistent with their observation: these filaments are easier to fragment, but thermal fluctuations are not enough (over our time scales, at least) and additional factors are required, such as mechanical stress (pipetting filament solutions) or severing proteins (cofilin). The strong illumination of fluorescently labelled filaments is also known to induce severing. We discuss these aspects in the discussion section of our manuscript (p. 21).

As reported by Hung et al. Science 2011 (Supp Mat and Supp Fig S1), MICAL does not alter

actin dynamics through H₂O₂ production. To confirm this point, we monitored the barbed end depolymerization of ADP-F-actin in the presence of 40 mM H₂O₂, and found no significant difference with depolymerization in standard buffer. We now report these results in our new Fig. S7e.

We verified the potency of our H₂O₂ by monitoring the elongation of filaments from 1 μM G-actin incubated with 40 mM H₂O₂ for 20 minutes at 20°C and found that, in agreement with previous reports (DalleDonne et al. Biophys J 1995), H₂O₂ oxidizes G-actin and hinders polymerization: filaments elongated from H₂O₂-treated monomers at 2.4 ± 0.3 sub/s (Std Dev, N=12), compared to 7.8 ± 0.8 sub/s (Std Dev, N=12) in control experiments. This has been added in the Figure legend of new Fig. S7e.

Minor:

1. Labels too small in some of the figures.
2. Correct IR1055E in the legend to Figure S2.

These issues have been addressed. In particular, the size of the labels has been increased in Fig. 1 (a-c), Fig. 2 (a, c, d, e), Fig. 3 (b-d), Fig. 4 (a-i), Fig. 6e, Fig. S1 (d, g), Fig. S6 (a-b).

Reviewer #3 (Remarks to the Author, final requests):

This review is an update on my previous review for NCB. The authors have addressed most of my concerns. The paper appears appropriate for Nature Communications. It must be pointed out, however, that some of the work presented here is not entirely novel. This includes the structural work, and well as the involvement of MICAL isoforms in cytokinesis. Thus, Liu et al., JBC 2016 already found that MICAL3 is involved in cytokinesis, and Rai et al., eLIFE 2016 determined a series of structures, including a complex of the C-terminal region of MICAL1 with Rab10. Nevertheless, the authors acknowledge these studies, and the work presented here implicates a different MICAL isoform in cytokinesis (MICAL1), and is more complete in that it addresses this topic using a comprehensive, cellular, biochemical and structural approach. The findings are important, and concern a fundamental biological process, which is of interest to the general readership of Nature Communications. A minor criticism -the paper would benefit from a more detailed discussion of MICAL function, including its N-terminal domain for which several structures have been determined (not referenced). In other words, how do the non-catalytic domains influence/regulate MICAL's activities?

We are pleased that the Reviewer is satisfied by the new data added in the revised manuscript.

For the final version of the manuscript, we have taken into account all editorial changes requested by the Reviewer:

References regarding the structure of the monoxygenase domain (ref. 26/27) and the CH domain (ref. 56) have been added in the introduction and in the discussion (p. 22). A discussion of the role of the non-catalytic domains has been added in the discussion (p. 22) together with the finding of the release of the intramolecular inhibitory conformation by Rab35:

“In vitro, the catalytic activity of MICAL1 monoxygenase domain)^{26, 27} is modulated by its non-catalytic CH, LIM and C-terminal domains (ref.^{24, 56} and see below).”